



# Quantification of lightning-produced $NO_x$ over the Pyrenees and the Ebro Valley by using different TROPOMI-$NO_2$ and cloud research products

Francisco J. Pérez-Invernón[1], Heidi Huntrieser[1], Thilo Erbertseder[2], Diego Loyola[3], Pieter Valks[3], Song Liu[3], Dale Allen[4], Kenneth Pickering[4], Eric Bucsela[5], Patrick Jöckel[1], Jos van Geffen[6], Henk Eskes[6], Sergio Soler[7], Francisco J. Gordillo-Vázquez[7], and Jeff Lapierre[8]

[1]Deutsches Zentrum für Luft- und Raumfahrt, Institut für Physik der Atmosphäre, Oberpfaffenhofen, Germany
[2]Deutsches Zentrum für Luft- und Raumfahrt, Deutsches Fernerkundungsdatenzentrum, Oberpfaffenhofen, Germany
[3]Deutsches Zentrum für Luft- und Raumfahrt, Methodik der Fernerkundung, Oberpfaffenhofen, Germany
[4]University of Maryland, USA
[5]SRI International, USA
[6]Royal Netherlands Meteorological Institute, Netherlands
[7]Instituto de Astrofísica de Andalucía, CSIC, Glorieta de la Astronomía s/n, Granada, Spain
[8]Earth Networks, Germantown, MD, USA

**Correspondence:** Francisco J. Pérez-Invernón (FranciscoJavier.Perez-Invernon@dlr.de)

**Abstract.** Lightning is one of the major sources of nitrogen oxides ($NO_x$) in the atmosphere, contributing to the tropospheric concentration of ozone and to the oxidising capacity of the atmosphere. Lightning produces between 2-8 Tg N per year globally and on average about $250 \pm 150$ mol $NO_x$ per flash. In this work, we estimate the moles of $NO_x$ produced per flash ($LNO_x$ production efficiency) in the Pyrenees (Spain, France and Andorra) and in the Ebro Valley (Spain) by using nitrogen dioxide
5 ($NO_2$) and cloud properties from the TROPOspheric Monitoring Instrument (TROPOMI) and lightning data from the Earth Networks Global Lightning Network (ENGLN) and from the EUropean Co-operation for LIghtning Detection (EUCLID). The Pyrenees is one of the areas in Europe with the highest lightning frequency and, due to its remoteness as well as experiencing very low $NO_x$ background, enables us to better distinguish the $LNO_x$ signal produced by recent lightning in TROPOMI $NO_2$ measurements. We compare the $LNO_x$ production efficiency estimates for 8 convective systems in 2018 using two different
10 sets of TROPOMI research products, provided by the Royal Netherlands Meteorological Institute (KNMI) and the Deutsches Zentrum für Luft- und Raumfahrt (DLR), respectively. According to our results, the mean $LNO_x$ production efficiency in the Pyrenees and in the Ebro Valley, using a three-hour chemical lifetime, ranges between 14 and 103 mol $NO_x$ per flash from the 8 systems. The mean $LNO_x$ production efficiency estimates obtained using both TROPOMI products and ENGLN lightning data differ by $\sim$23%, while it differs by $\sim$35% when using EUCLID lightning data. The main sources of uncertainty when
15 using ENGLN lightning data are the estimation of background $NO_x$ that is not produced by lightning and the time window before the TROPOMI overpass that is used to count the total number of lightning flashes contributing to fresh-produced $LNO_x$. The main source of uncertainty when using EUCLID lightning data is the uncertainty in the detection efficiency of EUCLID.



# 1 Introduction

Lightning is one of the major sources of nitrogen oxides ($NO_x$ = NO + $NO_2$) in the upper troposphere [e. g., Schumann and
Huntrieser (2007) and references therein]. Lightning channels are formed by plasma reaching several thousands of Kelvin
(Wallace, 1964). Such a high temperature produces dissociation of nitrogen and oxygen air molecules (Ripoll et al., 2014b,
a; Kieu et al., 2021), contributing to the formation of $NO_x$ by the Zeldovich mechanism (Zeldovich et al., 1947). Lightning-
induced nitrogen oxides ($LNO_x$) contribute about 10% to global $NO_x$ emissions and play an important role in determining the
concentration of ozone and other chemical species in the upper troposphere as well as the oxidising capacity of the atmosphere
(e.g., Labrador et al., 2005; Schumann and Huntrieser, 2007; Murray et al., 2012; Gordillo-Vázquez et al., 2019). Lightning
produces between 2-8 Tg N per year globally (100-400 mol $NO_x$ per flash) and on average about 250 mol $NO_x$ per flash
(Schumann and Huntrieser, 2007) .

Reducing the uncertainty of the $NO_x$ production by lightning and understanding the factors that influence this production
is still a challenge. Aircraft measurements have significantly contributed to determining the production of $NO_x$ per flash, or
$LNO_x$ Production Efficiency (PE) (e.g., Huntrieser et al., 2002, 2016; Allen et al., 2021b). However, aircraft campaigns cannot
provide a continuous monitoring of $LNO_x$ and are difficult to carry out in some regions. Nadir-viewing satellite instruments
such as the Ozone Monitoring Instrument (OMI), the Scanning Imaging Absorption spectroMeterfor Atmospheric CHartogra-
phY (SCIAMACHY) and the TROPOspheric Monitoring Instrument (TROPOMI) estimate the column densities of $NO_2$ over
thunderstorms. Several authors have used OMI $NO_2$ measurements to estima the the $LNO_x$ PE in a case-based approach or
systematically over different regions (Beirle et al., 2010; Marais et al., 2018), including midlatitude regions (Bucsela et al.,
2019), tropical regions (Allen et al., 2019) and the U.S. (e.g., Pickering et al., 2016; Lapierre et al., 2020; Zhang et al., 2020;
Allen et al., 2021a). Satellite-based measurements can help to estimate $LNO_x$ amounts over regions where aircraft campaigns
are rare or to systematically investigate possible relationships between the characteristics of thunderstorms and $LNO_x$ over
different geographical regions (Bucsela et al., 2019). However, the opacity of thunderclouds can strongly affect the retrieval of
$NO_2$ (Beirle et al., 2009), while convection can transport $NO_x$ released at the surface to the upper troposphere, where it is mixed
with freshly produced $LNO_x$. Therefore, the use of atmospheric and radiative models in combination with $NO_2$ measurements
is needed to estimate the $NO_x$ Production Efficiency ($LNO_x$ PE).

The TROPOMI instrument on board the European Space Agency Sentinel-5 Precursor (S5P) satellite was launched on 13
October 2017. TROPOMI operates from a low Earth polar orbit that provides daily global measurements of several trace
gases (including $NO_2$) and cloud properties (Veefkind et al., 2012). The horizontal resolution at nadir before 6 August 2019 is
3.6 km × 7.2 km, while it is 3.6 km × 5.6 km thereafter. This unprecedented spatial resolution represents a unique opportunity
to investigate the $LNO_x$ PE from satellite measurements. Recently, Allen et al. (2021a) used, for the first time, TROPOMI
measurements to estimate the $LNO_x$ PE for 29 cases in the USA using lightning data from the Earth Network Global Lightning
Network (ENGLN) and from the Geostationary Lightning Mapper (GLM) aboard the Geostationary Operational Environmental
Satellite-16 (GOES-16). They reported 175 ± 100 and 120 ± 65 mol $NO_x$ per flash using ENGLN and GLM lightning data,



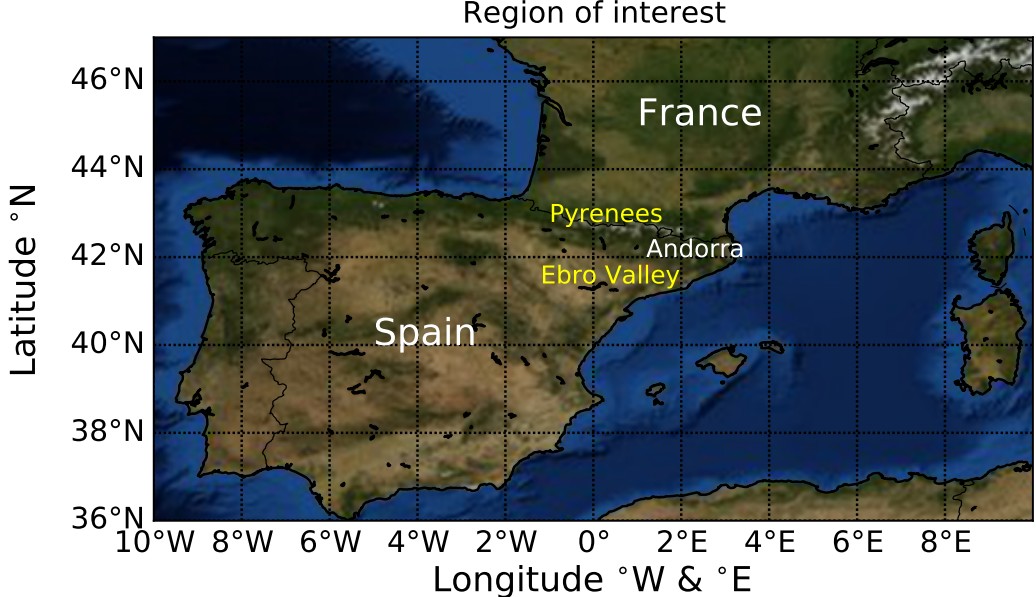

**Figure 1.** Geographical location of the region of interest (Pyrenees and Ebro Valley). The map has been extracted from Blue Marble images provided by NASA (National Aeronautics and Space Administration, 2021).

respectively. These values are at the lower end of the globally averaged $LNO_x$ PE of $250 \pm 150$ mol $NO_x$ per flash as given by Schumann and Huntrieser (2007).

In this work, we, for the first time, quantify the amount of $LNO_x$ over the Pyrenees and the Ebro Valley in Spain by using different TROPOMI-$NO_2$ and cloud research products provided by two different European research institutes, such as KNMI
and DLR. The geographical location of the Pyrenees and the Ebro Valley is indicated in Figure 1. The Pyrenees are one of the areas in Europe with the highest lightning frequency (Molinie et al., 1999; Pineda et al., 2010; Anderson and Klugmann, 2014) and is a good place to distinguish the $LNO_x$ signal due to its remoteness and very low $NO_x$ background (Vinken et al., 2014). Airflows over the studied areas are influenced by the proximity of the Mediterranean Sea and the Atlantic Ocean, the high mountains of the Pyrenees, cold fronts cossing Europe, and thermal low centered over the Iberian Peninsula (Pineda et al.,
2010). In this study, we analyze 8 thunderstorms taking place in April and May 2018, the months with the highest occurrence of lightning in Spain (Pineda et al., 2010). During late spring, lightning activity in the area reaches its maximum over the mountains and is driven by solar heating (Esteban et al., 2006; Pineda et al., 2010). Therefore, we expect that during this time of the year a number of thunderstorms are active during the TROPOMI overpass ($\sim$13:30 LT). We combine two TROPOMI research products with lightning data from the ENGLN (Zhu et al., 2017; Lapierre et al., 2020) and the EUropean Co-operation
for LIghtning Detection (EUCLID) systems (Schulz et al., 2016). Apart from providing new valuable estimates of $LNO_x$ for Europe, this analysis will enable us to quantify the influence of using different lightning data sets and different TROPOMI $NO_2$ and cloud research products for the estimates of $LNO_x$ PE.



## 2 Data sets and methods

### 2.1 TROPOMI NO$_2$ and cloud research products

We use TROPOMI NO$_2$ and cloud research products for 8 deep convective systems in the Pyrenees between April and May 2018. TROPOMI is a passive imaging spectrometer with 8 spectral bands covering the ultraviolet (UV), visible (VIS), near infrared (NIR), and short-wavelength IR (SWIR) spectral regions (Veefkind et al., 2012). TROPOMI provides spectral data that is combined with different methods/algorithms to retrieve NO$_2$ concentrations and cloud properties (e.g., Wang et al., 2008; Loyola et al., 2018; Marais et al., 2021; Liu et al., 2021). In this work, we use two different sets of TROPOMI research

products. The variables extracted from the TROPOMI products are the Slant Column Density (SCD) NO$_2$, the error of the SCD NO$_2$, the quality assurance (QA) value, the stratospheric Vertical Column Density (VCD) of NO$_2$, the stratospheric Air Mass Factor (AMF), the Cloud Fraction (CF) and the Optical Centroid Pressure (OCP).

  The first set of TROPOMI research product is here referred to as the Royal Netherlands Meteorological Institute (KNMI) version 2.1 research product (Allen et al., 2021a) (TROP-KNMI) based on the official TROPOMI NO$_2$ Algorithm Theoretical

Basis Document (ATBD) (van Geffen et al., 2021). The TROP-KNMI cloud research product is based on the Fast Retrieval Scheme for Clouds from the Oxygen A-band-S (FRESCO-S) algorithm with a Cloud as Reflecting Boundaries (CRB) model of clouds (Koelemeijer et al., 2001). In the CRB model, clouds are described as a Lambertian reflecting boundary. The separation of the contribution of the troposphere and stratosphere to the NO$_2$ column density for the TROP-KNMI NO$_2$ research product is based on a priori chemical profiles from the chemistry transport model TM5-MP (Myriokefalitakis et al., 2020).

We use the version 2.1_test of this product, a modified NO$_2$ product that increases the data coverage over bright pixels over deep convective clouds and includes spike removal to better deal with saturation and blooming effects in the radiance spectra (Ludewig et al., 2020; Allen et al., 2021a). The reflectance value at 440 nm is reconstructed from the Differential Optical Absorption Spectroscopy (DOAS) method polynomial and the Ring correction as input to the routine that calculates the cloud (radiance) fraction in the NO$_2$ window. We refer to van Geffen et al. (2021); Allen et al. (2021a) for a detailed description

of the TROP-KNMI NO$_2$ and cloud research products. Following Allen et al. (2021a), we use pixels with a quality assurance value above 0.28 (fair quality). This selection ensures that the SCD NO$_2$ error is less than 2 petamolec cm$^{-2}$.

  We refer to the second set of TROPOMI research product as the Deutsches Zentrum für Luft- und Raumfahrt (DLR) research product (TROP-DLR). The TROP-DLR cloud research product uses the OCRA/ROCINN algorithms for retrieving cloud properties (Loyola et al., 2018). The cloud properties provided by ROCINN uses the Clouds-As-Layers (CAL) model (Loyola et al.,

2018). In the CAL model, clouds are treated as optically uniform layers using a more realistic cloud scattering model than the CRB model (Lindfors et al., 2018). We refer to Loyola et al. (2018) for a more extended description of the TROP-DLR cloud research product. The TROP-DLR NO$_2$ research product uses a Directionally dependent STRatospheric Estimation Algorithm from Mainz (DSTREAM) to separate the contribution of the troposphere and stratosphere to the NO$_2$ column density (Liu et al., 2021). This method does not require any input from atmospheric models. The STREAM method does not dis-

tinguish free tropospheric diffuse NO$_2$ from stratospheric NO$_2$. This is different in the TROP-KNMI approach, where a free tropospheric column is derived from the TM5-MP profiles. In the case of TROP-KNMI, stratospheric NO$_2$ retrieval does not





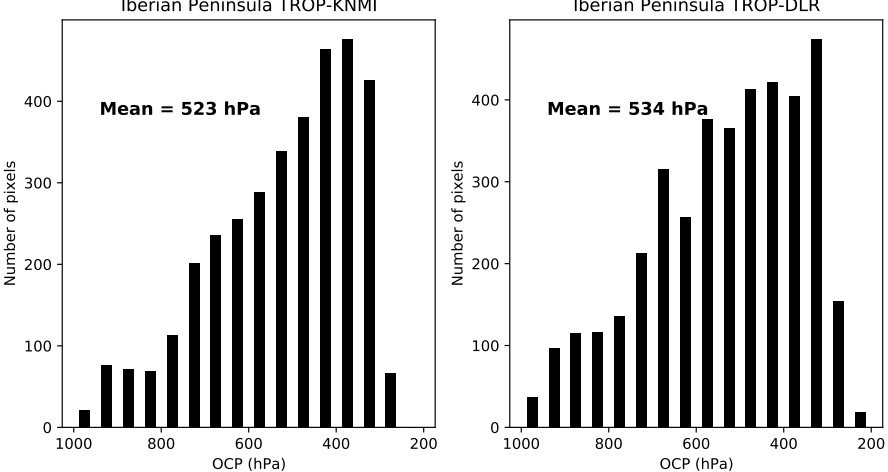

**Figure 2.** Distributions of OCP for pixels containing ENGLN flashes for the TROP-KNMI (left panel) and the TROP-DLR (right panel) products for all the studied cases.

include free tropospheric $NO_2$, while it does include free tropospheric $NO_2$ in the case of the TROP-DLR product. So, we expect the tropospheric backgrounds to be substantially higher in the TROP-KNMI product than for the TROP-DLR product. The detailed description of the TROP-DLR $NO_2$ research product can be found in Liu et al. (2021). In this work, we use pixels
with a SCD $NO_2$ error lower than 2 petamolec cm$^{-2}$ to be consistent with the QA threshold defined for the TROP-KNMI product.

     Pixels with deep convection are defined as pixels in which the effective cloud fraction is greater than 0.95 (Allen et al., 2021a) and the OCP value is lower than a threshold. The threshold is defined as the averaged OCP for all lightning flashes included in this study. We calculate it using OCP values for every pixels containing lightning flashes according to the TROPOMI cloud
products, providing that the OCP value is not undefined. The averaged OCP for the TROP-KNMI and the TROP-DLR products are 523 hPa and 534 hPa, respectively. These pressures are slightly higher than the 500 hPa threshold employed by Pickering et al. (2016) and Allen et al. (2021a) for deep convective systems over the USA. Figure 2 shows the distributions of OCP values for TROP-KNMI and TROP-DLR using ENGLN lightning data over all the studied cases. Both distributions peak around 400 hPa, while there are more lightning flashes taking place in pixels with OCP values between 650 hPa and 500 hPa
in the case of the TROP-DLR product than the TROP-KNMI product (3923 versus 3489 pixels). We have calculated the T-test for the means of the OCP distributions plotted in Figure 2, obtaining a p-value lower than 0.05. This p-value indicates that differences in the mean OCP derived from the TROP-KNMI and the TROP-DLR products are statistically significant.





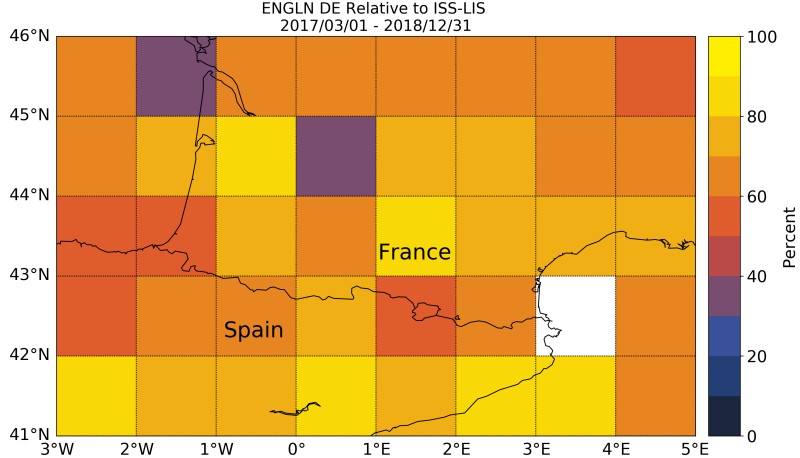

**Figure 3.** Spatial distribution of the ENGLN DE (in %) relative to ISS-LIS between March 2017 and December 2018 over Northern Spain, Southern France and Andorra.

## 2.2 Lightning measurements

We use lightning data provided by two lightning location systems, ENGLN and EUCLID, to calculate the amount of $LNO_x$
produced per flash (or $LNO_x$ PE).

The ENGLN is a global network composed of both broadband sensors from the Earth Networks Total Lightning Network (Liu et al., 2014) and Very Low Frequency (VLF) sensors from the World Wide Lightning Location Network (Hutchins et al., 2012) that provide the position, time of occurrence, polarity and peak current of lightning strokes. ENGLN has a Detection Efficiency (DE) of about 90% for Cloud-to-Ground (CG) strokes over the USA (Marchand et al., 2019). In this work, we use
the flash product provided by ENGLN. This product is based on the flash criteria proposed by Liu and Heckman (2011), to cluster these strokes into flashes, in which two strokes are part of the same flash if they occur in a 0.7 s temporal window and in a 10 km spatial window.

We use lightning data from the Lightning Imaging Sensor (LIS) onboard the International Space Station (ISS) (Blakeslee et al., 2020) to estimate the DE of ENGLN over the Pyrenees. ISS-LIS detects optical emissions from lightning with a frame
integration time of 1.79 ms with a spatial resolution of 4 km (Bitzer and Christian, 2015; Blakeslee et al., 2020). LIS assorts contiguous events into groups, and clusters groups into flashes with a temporal criteria of 330 ms and a spatial criteria of 5.5 km (Mach et al., 2007). We compare ENGLN and ISS-LIS lightning data over the Pyrenees using the Bayesian approach proposed by Bitzer et al. (2016) with 330 ms and 25 km as the matching criteria. The Bayesian approach is more accurate than direct comparison between lightning data, as neither of the detection systems can be characterized as the truth. We show in Figure 3
the spatial distribution of the obtained ENGLN DE over the Pyrenees. The average DE in this region is $68 \pm 12\%$.



EUCLID is a European network composed of 149 lightning sensors manufactured by Vaisala Inc. and distributed over Europe (Schulz et al., 2016). Despite the high DE of EUCLID over Europe, the mean DE of EUCLID over the Pyrenees and the Ebro Valley is only about 30-60% (Poelman and Schulz) because of the low number of stations over that area and in Africa. We have selected two thunderstorms taking place between April and May 2018 over the Pyrenees and the Ebro Valley that were

simultaneously detected by EUCLID and ISS-LIS. We have compared the total number of flashes reported by EUCLID and ISS-LIS in both thunderstorms, calculating a DE of 0.40 in the Pyrenees and a DE of 0.15 in the Ebro Valley. We use $27\% \pm 12\%$ as the DE correction for EUCLID. The significant difference between the DE of EUCLID and ENGLN over the Pyrenees represents a good opportunity to investigate the influence of Lightning Location Systems (LSS) DE on the $LNO_x$ PE.

### 2.3   Meteorological and chemistry data

As we will describe in section 2.4, estimating the tropospheric background concentration of $NO_x$ ($NO_x$ that is not produced by lightning) is essential for the calculation of $LNO_x$. Although the Pyrenees is an area with relatively low background-$NO_x$ concentration (Vinken et al., 2014), tropospheric background-$NO_x$ can be transported from the boundary layer to the upper troposphere by convection or advected from the Ebro Valley or the city of Barcelona. Therefore, we cannot neglect the background-$NO_x$ and have to subtract it from the VCD satellite measurements. To account for this, we use a combination of

meteorological and chemical data as described below.

We use meteorological data provided by the European Centre for Medium-Range Weather Forecasts (ECMWF) ERA5-reanalysis data set. In this work, we use the 1-hourly ERA5 horizontal wind averaged between 200 hPa and 500 hPa pressure levels with a horizontal resolution of 0.25°. For each TROPOMI pixel containing lightning flashes prior to the TROPOMI overpass, we use the wind velocity and direction to estimate the advection of $LNO_x$. All the pixels that satisfy the deep

convection constraint and that are not influenced by the spreading of $LNO_x$, are then considered as non-flashing pixels and employed to estimate the background-$NO_x$.

Alternatively, we use airborne measurements to estimate the background-NO. In particular, we use NO measurements from the In-service Aircraft for a Global Observing System (IAGOS) and from the Civil Aircraft for the Regular Investigation of the Atmosphere Based on an Instrument Container (CARIBIC) NO measurements (Brenninkmeijer et al., 2007). On 22 June,

2005, a CARIBIC flight passed over a convective system in the Pyrenees. Unfortunately, we do not have access to lightning data for that day, only cloud satellite products. However, the measured ratio $NO/NO_y$ can be used to estimate the age of the freshly produced $NO_x$ (Huntrieser et al., 2002). The measured ratio of NO to $NO_y$ (about 0.1) during the passage over the convective system suggests no impact of fresh $LNO_x$. The measured mixing ratio of CO can be used as a proxy for upward transport of NO from the boundary layer (Huntrieser et al., 2002). Measured simultaneous increases of CO and NO on 22 June, 2005 flight

suggest upward transport of polluted boundary layer air, confirming that the airplane passed across a convective system. The measured mixing ratio of NO at 12 km altitude during the passage over the convective system was $0.3 \pm 0.1$ ppb, in agreement with previous airborne NO measurements over convective systems without lightning in Europe during the EULINOX campaign (Huntrieser et al., 2002). We assume a NO/NO2 ratio in the upper troposphere of 2 mol mol$^{-1}$ (Silvern et al., 2018). Therefore,





we use 0.45 ppb as an alternative to the estimation of the background-$NO_x$ from non-flashing pixels. The method we used to transform this mixing ratio of $NO_x$ into petamolec cm$^{-2}$ is described in more detail in Section 2.4.

We can estimate the VCD of $NO_x$ using CARIBIC measurement at 12 km. We assume that the shape of the vertical profile of $NO_x$ of the 22 June, 2005 convective system case is similar to the mean vertical profile of $NO_x$ reported by Huntrieser et al. (2002) in Europe (Fig. 7a in (Huntrieser et al., 2002)). Using the shape of the EULINOX profile and the CARIBIC measurement at 12 km, we can estimate the mixing ratio of $NO_x$ between the surface and 12 km level. Finally, we can integrate the vertical profile to obtain the VCD of $NO_x$, resulting in 0.75 petamolec cm$^{-2}$.

## 2.4 Calculation of the $LNO_x$ Air Mass Factor

TROPOMI provides SCD $NO_2$ over the cloud top and in the upper parts of the clouds. As we will see in section 2.5, our $LNO_x$ PE algorithm requires the VCD $LNO_x$ to be determined from the SCD $NO_2$. The ratio to convert SCD $NO_2$ to VCD $LNO_x$ is called the $AMF_{LNOx}$ and its calculation requires a priori estimations of the mean $LNO_2$ and $LNO_x$ profiles over the studied region (Pickering et al., 2016) and of the absorption of the atmosphere (Beirle et al., 2009; Bucsela et al., 2013).

We employ the ECMWF – Hamburg (ECHAM)/Modular Earth Submodel System (MESSy version 2.54.0) Atmospheric Chemistry (EMAC) model (Jöckel et al., 2016) to extract the mean $LNO_2$ and $LNO_x$ profiles over the studied area by performing two simulations (with and without lightning). We perform the simulations following the Quasi Chemistry-Transport Model (QCTM) mode proposed by Deckert et al. (2011). Firstly, we perform a one year global simulation (January 1, 2018 to January 1, 2019) without lightning nudged towards ERA-Interim reanalysis meteorological fields. Secondly, we perform a second simulation with lightning for the same period using numerically identical meteorological fields as the simulation without lightning. The QCTM mode decouples the dynamics from the chemistry in order to operate the model as a chemistry-transport model, implying that small chemical perturbations do not alter the simulated meteorology by introducing noise (Deckert et al., 2011). The simulations are conducted in T42L90MA resolution, i.e. with a quadratic Gaussian grid of 2.8°×2.8° in latitude and longitude with 90 vertical levels reaching up to the 0.01 hPa pressure level and with 720 s time steps (Jöckel et al., 2016). $LNO_x$ is calculated by using the MESSy submodel LNOX (Tost et al., 2007). Lightning is parameterized according to the updraft velocity (Grewe et al., 2001) and using a scaling factor that ensures a global lightning occurrence rate of ∼45 flashes per second (Christian et al., 2003; Cecil et al., 2014). We set the production of $NO_x$ per flash following Price et al. (1997) and employ the C-shaped vertical profiles of $LNO_x$ reported by Pickering et al. (1998). We use the same chemical setup and chemical mechanism as described by (Jöckel et al., 2016) for RC1 simulations.

We extract the vertical profiles of NO and $NO_2$ with and without lightning for May 2018 coincident with the TROPOMI overpass time to calculate the $LNO_2$ and $LNO_x$ vertical profiles. We obtain that the day in May 2018 with the highest $LNO_x$ column density is May 13, 2018. Figure 4 shows the vertical profiles obtained from the EMAC simulations. Both $LNO_x$ and $LNO_2$ vertical profiles peak between 300 hPa and 250 hPa pressure levels (between ∼9 and 11 km altitude), while the vertical profiles of $LNO_x$ and $LNO_2$ calculated by Pickering et al. (2016) over the Gulf of Mexico peak at about 150 hPa. The reason for this difference is that thunderstorms are taller at sub-tropical latitudes than at mid-latitudes. Non-negligible values of $LNO_x$ and $LNO_2$ values between 100 and 200 hPa (Figure 4) may have been transported to the Pyrenees from tropical latitudes.

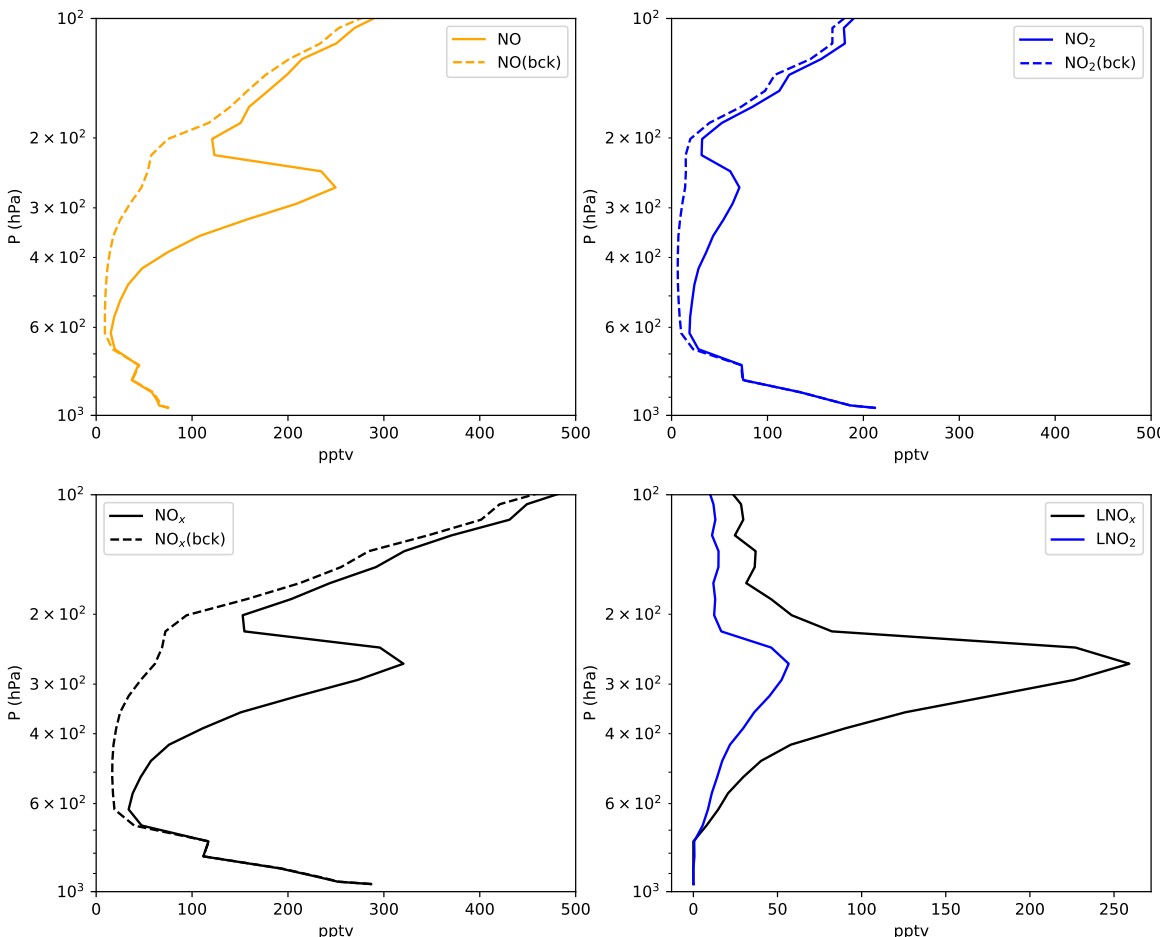

**Figure 4.** Vertical mixing ratio profiles of NO (upper left panel), $NO_2$ (upper right panel), $NO_x$ (lower left pannel), $LNO_x$ and $LNO_2$ (lower right pannel) extracted from EMAC simulations with (solid lines) and without (dashed lines) lightning (background: bck) on 13 May, 2018 at 12 h LT (close to the TROPOMI overpass).

We use the $LNO_2$ and $LNO_x$ vertical profiles from the simulations to calculate the $AMF_{LNOx}$ following Bucsela et al. (2013). We use the TOMRAD forward vector radiative transfer model (Dave, 1965) to calculate the scattering weights for each of the 8 studied cases using the viewing geometry and the cloud properties for each pixel. We obtained $AMF_{LNOx}$ values ranging between 0.28 and 0.71.



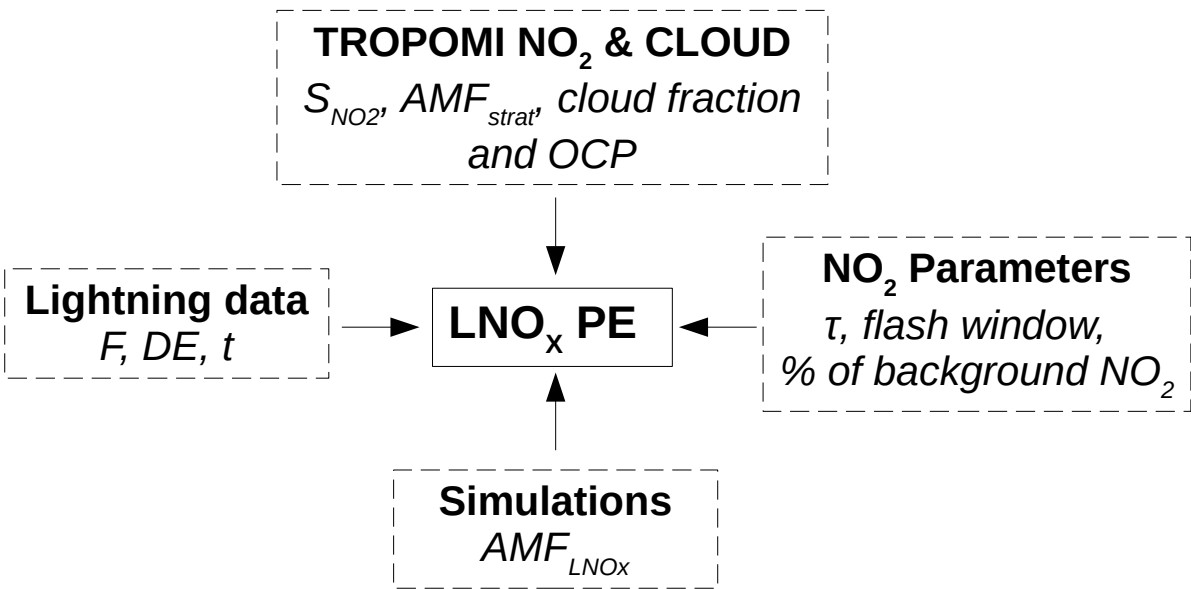

**Figure 5.** Overview graphic showing the variables that are included in the calculation of $LNO_x$ PE.

## 2.5 Calculation of the $LNO_x$ PE

We use the TROPOMI $LNO_x$ PE method proposed by Allen et al. (2021a). Figure 5 shows an overview graphic indicating the variables that are included in the calculation of $LNO_x$ PE, while Appendix 5 indicates the list of acronyms. The source of these variables are TROPOMI products, lightning data, simulations and parameters that are introduced based on literature. The $LNO_x$ PE is calculated as

$$PE = [V_{tropLNOx} \times A] / \left[ N_A \times DE^{-1} \sum_i (F \times \exp(-t_i/\tau)) \right], \tag{1}$$

where $PE$ are the moles of $NO_x$ produced per flash, $V_{tropLNOx}$ is the tropospheric column of $NO_x$ produced by recent lightning (molec cm$^{-2}$), $A$ is the area (cm$^{-2}$) of the thunderstorm with deep convection or with undefined OCP, $N_A$ is the Avogadro's number (molec mol$^{-1}$), DE is the detection efficiency of ENGLN or EUCLID, $\tau$ is the lifetime of $NO_x$ in the near field of convection, assumed as 3 hours (Nault et al., 2017; Allen et al., 2021a), $t_i$ is the age of individual flashes at the time of the overpass (the time since the flash occurred) and $F$ is the total number of flashes 5 hours prior to the TROPOMI overpass





of each pixel. We use a 5 h flash window because it is larger than the assumed 3 hours lifetime of $NO_x$ in the near field of convection. Sensitivity studies using other flash windows are performed in Section 3.3.

$V_{tropLNOx}$ is calculated as

$$V_{tropLNOx} = Median(V_{tropNOx}) - V_{tropbck}, \tag{2}$$

where $V_{tropNOx}$ is the VCD $NO_x$ over pixels with deep convection or with undefined cloud fraction and $V_{tropbck}$ is the background-$NO_x$. We use the median instead of the mean of $V_{tropNOx}$ in order to remove the influence of possible outlier pixels. $V_{tropNOx}$ is defined as

$$V_{tropNOx} = [S_{NO2} - avg(V_{stratNO2} \times AMF_{strat})] / AMF_{LNOx}, \tag{3}$$

where $S_{NO2}$ is the SCD of $NO_2$, $V_{stratNO2}$ is the stratospheric VCD of $NO_2$ and $AMF_{strat}$ is the stratospheric AMF.

Following Allen et al. (2021a), we calculate $V_{tropbck}$ as the $30^{th}$ and the $10^{th}$ percentile of $V_{tropNOx}$ over non-flashing pixels with deep convection. These percentiles are in agreement with airborne measurements during the EULINOX campaign (Huntrieser et al., 2002). Alternatively, we calculate the background as the mean $V_{tropNOx}$ concentration averaged over three

days with low lightning activity over the Pyrenees from TROPOMI data and using CARIBIC measurements in a convective system with low lightning activity over the Pyrenees (as described in section 2.3).

## 2.6 Calculation of the background-$NO_x$ based on days with low lightning activity

Apart from calculating the background-$NO_x$ from non-flashing pixels in a case-based approach, we have selected three cases with low lightning activity before the TROPOMI overpass to estimate the mean background-$NO_x$ over convective systems.

In particular, we have used TROPOMI measurements on 8 April, 12 April and 13 April 2018 in the region between 41°N - 45°N degrees latitude and 3°W - 5°E degrees longitude. The total number of lightning flashes prior to the TROPOMI overpass for the three studied cases were, respectively, 149, 65 and 50. The mean $V_{tropNOx}$ during these days using the TROP-KNMI research product were 1.07 petamolec cm$^{-2}$, 1.98 petamolec cm$^{-2}$ and 0.39 petamolec cm$^{-2}$, while the $V_{tropNOx}$ using the TROP-DLR research product were 0.37 petamolec cm$^{-2}$, 1.00 petamolec cm$^{-2}$ and -0.5 petamolec cm$^{-2}$. Negative values

suggest that the average stratospheric column exceeding the local vertical column (eq. (3)) or the tropospheric background exceeding the signal (eq. (2)). The average background $V_{tropNOx}$ for the TROP-KNMI and the TROP-DLR research products were, respectively, 1.06 petamolec cm$^{-2}$, and 0.37 petamolec cm$^{-2}$. These estimates are, respectively, slightly above and below the background VCD of $NO_x$ estimated using CARIBIC measurements (0.75 petamolec cm$^{-2}$).





## 3 Results

In this section we present LNO$_x$ estimates for 8 selected cases. We describe the TROPOMI product for the selected cases in Section 3.1. The LNO$_x$ PE estimates are presented in Sections 3.2, while a sensitivity analysis of the results is discussed in Section 3.3.

### 3.1   Selected case studies

The 8 selected cases correspond to 8 thunderstorms that were active no more than 5 hours before the TROPOMI overpass
on the following days: 29 April, 7 May, 12 May, 21 May, 22 May, 26 May, 28 May and 30 May 2018. Unfortunately, the TROP-DLR research product was not available for the case on 30 May 2018. In addition, the thunderstorm taking place on 26 May 2018 had a significant lightning activity between 45°N and 46°N, but we do not have access to EUCLID data north of 45°N.

Figure 6 shows the ENGLN lightning data and some of the variables from the TROP-DLR product for the case 29 April 2018.
Figure 7 is similar as Figure 6 but instead showing EUCLID lightning data and some of the variables from the TROP-KNMI product. Lightning activity is distributed between the Ebro Valley, the Pyrenees and the French coast.

The upper left panels of Figures 6 and 7 show the position of lightning flashes and the calculated VCD NO$_x$ in pixels with deep convection. A comparison of the upper left of Figures 6 and 7 shows that there are more lightning flashes reported by ENGLN than by EUCLID. The upper right panels show the SCD-NO$_2$ for each of the used TROPOMI products, indicating that
there are not significant differences between them. Areas with high lightning activity coincide with areas with high SCD-NO$_2$, suggesting that the LNO$_x$ signal is detectable by TROPOMI. There are also high SCD-NO$_2$ values near the city of Barcelona, a highly populated area producing high emissions of NO$_x$.

The center left and right panels show the stratospheric VCD of NO$_2$ and the stratospheric AMF of NO$_2$, respectively. The VCD$_{stratNO2}$ from the TROP-DLR product is slightly larger than from the TROP-KNMI product, while both the stratospheric
VCD of NO$_2$ and the stratospheric AMF of NO$_2$ are more homogeneous for the TROP-DLR product than for the TROP-KNMI product. The method to separate the contribution of the troposphere and stratosphere to the NO$_2$ column density are different for each product, which can affect the spatial distribution of the VCD$_{stratNO2}$ and the AMF$_{stratNO2}$. The TROP-KNMI NO$_2$ product uses a priori chemical profiles from the chemistry transport model TM5-MP (Myriokefalitakis et al., 2020), while the TROP-DLR NO$_2$ product uses the DSTREAM method to separate the contribution of the troposphere and stratosphere to the
NO$_2$ column density (Liu et al., 2021), (see section 2). Inhomogeneities in the TROP-KNMI product are due to jumps in the tropopause level. The TROP-KNMI product uses the temperature of the tropopause, which may jump up and down by a few levels linked to horizontal changes in temperature gradients. The STREAM model used in the TROP-DLR product will absorb free tropospheric NO$_2$ into the stratosphere, while the free tropospheric background may be overestimated in the TM5-MP model which is used to estimate the stratospheric column in the TROP-KNMI product.

Finally, the lower panels show that there are not significant differences between the cloud products, except for some pixels in which the TROP-DLR product estimates larger cloud fractions. The existence of more pixels with high cloud fractions in the





TROP-DLR product, than in the TROP-KNMI product, can influence the total number of pixels labeled as cloud convective pixels.

We present in Figures 8 and 9 similar plots for the case 7 May. As in the case 29 April, lightning activity is distributed
between the Ebro Valley, the Pyrenees and the French coast. Areas with high lightning activity coincide with areas with high SCD-NO$_2$, while there are also also high SCD-NO$_2$ values near the city of Barcelona. We can appreciate the same differences between the TROP-KNMI and the TROP-DLR products as in the case 29 April.

Figures 10 and 11 show plots for the case 28 May 2018. In this case, lightning activity is limited to the Ebro valley and the Pyrenees. There is a profuse LNO$_x$ signal in the SCD-NO$_2$ map. The stratospheric VCD of NO$_2$ and the stratospheric AMF
of NO$_2$ provided by the TROP-KNMI product are more homogeneous than in the previous two cases. The rest of the cases analyzed in this study are plotted in the Supplement.

Figure 12 shows the velocity and direction of the horizontal wind averaged between the 200 hPa and 500 hPa pressure levels for the cases on 29 April, 7 May and 28 May, 2018. On 29 April, 2018 strong southerly winds could have contributed to transport LNO$_x$ to the north, which is in agreement with the relative position of flashes and pixels with high concentration
of NO$_2$ as shown in Figures 7 and 6. On 7 May, 2018 northeasterly winds could have transported LNO$_x$ to the southwest according to the location of the flashes, in agreement with Figures 8, 9. Finally, the wind velocity was weak on 28 May, 2018 and transport of lightning NO$_x$ from the the flash positions is unlikely, in agreement with Figures 10 and 11.

### 3.2 LNO$_x$ PE estimates

In this section, we present the LNO$_x$ PE estimates for the selected cases using two different methods to estimate the background-
NO$_x$. The first method (subsection 3.2.1) is exclusively based on case by case TROPOMI measurements, as it uses non-flashing pixels with deep convection to estimate the background-NO$_x$. The second method (subsection 3.2.2) uses fixed values for the background-NO$_x$ from measurements over days with low lightning activity.

### 3.2.1 LNO$_x$ PE estimates using non-flashing pixels to estimate the background-NO$_x$

In this section, we present the LNO$_x$ PE estimates for the selected cases by using the $30^{th}$ and the $10^{th}$ percentile of $V_{tropNOx}$
over non-flashing pixels with deep convection as background-NO$_x$ estimations. Table 1 shows the results for 8 cases in the Pyrenees using the described method and the TROP-KNMI research product, while Table 2 shows the results using the TROP-DLR research product. Here we have used a 5 h time window before the TROPOMI overpass and a chemical lifetime of NO$_x$ ($\tau$) of 3 h for all the cases shown in these tables. We have chosen these values for the flash window and $\tau$ as reference values to show the LNO$_x$ estimates in Table 1. However, later in Section 3.3, we perform a sensitivity analysis using different values
for flash window and $\tau$.

Columns 1 and 2 show the date and thunderstorm region of each studied case and some mean values, respectively. Column 3 shows the total number of lightning flashes reported by ENGLN/EUCLID 5 h before the TROPOMI overpass without application of a DE. The total number of flashes reported by ENGLN is always larger than reported by EUCLID. Minor



**Table 1.** Results for the 8 studied cases in 2018 using the TROP-KNMI research product.

| Data | Region | F ENGLN/EUCLID (N flashes) | Mean OCP (hPa) | Median $V_{tropNOx}$ (petamolec cm$^{-2}$) | Mean $V_{stratNO2}$ (petamolec cm$^{-2}$) | Mean $AMF_{LNOx}$ | $V_{tropbck}$ $10^{th}/30^{th}$ (petamolec cm$^{-2}$) | PE (ENGLN) $30^{th}$ / $10^{th}$ mol NO$_x$/ f | PE (EUCLID) $30^{th}$ / $10^{th}$ mol NO$_x$/ f |
|---|---|---|---|---|---|---|---|---|---|
| 29 April | 40N-45N/3W-4E | 4591 / 982 | 628 | 3.8 | 7.5 | 0.72 | 2.7 / 3.1 | 22 / 42 | 34 / 72 |
| 7 May | 41N-44N/2W-4E | 5356 / 1044 | 346 | 3.4 | 6.9 | 0.36 | 1.3 / 2.0 | 30 / 47 | 81 / 124 |
| 12 May | 40N-45N/2W-2E | 1434 / 175 | 629 | 2.6 | 6.7 | 0.46 | 1.7 / 2.4 | 5 / 19 | 35 / 78 |
| 21 May | 42N-43.8N/2W-4E | 5263 / 1015 | 473 | 2.3 | 7.8 | 0.44 | 1.0 / 1.4 | 17 / 25 | 34 / 52 |
| 22 May | 41N-43N/1W-4E | 2318 / 515 | 530 | 2.6 | 7.8 | 0.46 | 1.6 / 1.8 | 19 / 26 | 32 / 46 |
| 26 May | 41N-46N/4W-2E | 25158 / 4821 | 593 | 6.4 | 7.2 | 0.34 | 2.8 / 3.4 | 86 / 103 | 42 / 54 |
| 28 May | 41N-43N/2W-4E | 7556 / 1568 | 494 | 5.2 | 5.7 | 0.45 | 3.5 / 3.9 | 52 / 72 | 99 / 139 |
| 30 May | 41N-45N/2W-4E | 9782 / 5754 | 502 | 1.8 | 8.9 | 0.80 | -0.01 / 0.8 | 65 / 115 | 83 / 102 |
| Mean $\pm \sigma$ | | | 527 | 3.5 | 7.3 | 0.50 | 1.8 / 2.3 | 47 $\pm$ 33 | 69 $\pm$ 34 |

**Table 2.** Results for the 7 studied cases in 2018 using the TROP-DLR research product.

| Data | Region | F ENGLN/EUCLID (N flashes) | Mean OCP (hPa) | Median $V_{tropNOx}$ (petamolec cm$^{-2}$) | Mean $V_{stratNO2}$ (petamolec cm$^{-2}$) | Mean $AMF_{LNOx}$ | $V_{tropbck}$ $10^{th}/30^{th}$ (petamolec cm$^{-2}$) | PE (ENGLN) $30^{th}$ / $10^{th}$ mol NO$_x$/ f | PE (EUCLID) $30^{th}$ / $10^{th}$ mol NO$_x$/ f |
|---|---|---|---|---|---|---|---|---|---|
| 29 April | 40N-45N/3W-4E | 4583 / 981 | 604 | 1.5 | 8.9 | 0.72 | 0.5 / 9.5 | 70 / 145 | 23 / 85 |
| 7 May | 41N-44N/2W-4E | 5241 / 1041 | 339 | 0.27 | 8.1 | 0.46 | -0.8 / -0.3 | 22 / 43 | 42 / 96 |
| 12 May | 40N-45N/2W-2E | 1409 / 171 | 573 | 0.89 | 8.0 | 0.59 | -0.8 / -0.3 | 40 / 78 | 40 / 62 |
| 21 May | 42N-43.8N/2W-4E | 5243 / 1012 | 440 | 0.89 | 8.4 | 0.54 | 0.05 / 0.5 | 38 / 62 | 37 / 47 |
| 22 May | 41N-43N/1W-4E | 2308 / 513 | 481 | 1.8 | 8.2 | 0.51 | 0.15 / 0.8 | 64 / 102 | 69 / 113 |
| 26 May | 41N-46N/4W-2E | 25233 / 4532 | 552 | 1.1 | 8.9 | 0.47 | -0.28 / 0.3 | 46 / 78 | 13 / 37 |
| 28 May | 41N-43N/2W-4E | 7543 / 1563 | 451 | 1.0 | 8.0 | 0.52 | -0.32 / 0.3 | 49 / 87 | 56 / 92 |
| Mean $\pm \sigma$ | | | 491 | 0.96 | 8.3 | 0.54 | -0.2 / 1.5 | 58 $\pm$ 33 | 51 $\pm$ 25 |

differences in the total number of flashes between both TROPOMI products (compare Tables 1 and 2) are due to minor
differences in the product grids.

Column 4 shows the OCP averaged for all lightning flashes reported by ENGLN. Significant differences are obtained between the cases. As lower limit, we obtain 339 hPa from the TROP-DLR research product for 7 May case, while we obtain
an upper limit of 629 hPa from the TROP-KNMI research product for the 12 May case. The mean OCP values for the TROP-KNMI and the TROP-DLR products are 527 hPa and 491 hPa, respectively. These values do not coincide with mean OCP
values showed in Fig. 2 because they correspond to the mean OCP per lightning flash instead of to the mean OCP value per
pixel. As a consequence, the mean OCP values showed in Column 4 are dominated by pixels with high lightning activity. The
OCP values depend on the intensity of convection in each thunderstorm as well as on the phase of the thunderstorm during the
TROPOMI overpass (Emersic et al., 2011).

Columns 5 and 6 of Tables 1 and 2 show the median tropospheric VCD of NO$_x$ and the mean stratospheric VCD of NO$_2$
($V_{tropNOx}$ and $V_{stratNO2}$) over pixels with deep convection, respectively. Higher values of $V_{stratNO2}$ for the TROP-DLR
research product compared to the TROP-KNMI product can be seen for all cases, except for the case on 30 May. As described





in section 2.5, $V_{tropNOx}$ is calculated by using a subtraction between the SCD of $NO_2$ and $V_{stratNO2}$. As $V_{stratNO2}$ is larger for the TROP-DLR research product, we receive lower values of $V_{tropNOx}$ than for the TROP-KNMI research product.

Column 7 shows the mean $AMF_{LNOx}$ over pixels with deep convection for each case. The value of $AMF_{LNOx}$ ranges be-
tween 0.34 and 0.80, while the averaged values for the TROP-KNMI and TROP-DLR products are 0.50 and 0.54, respectively. These values are in agreement with typical values reported by Allen et al. (2021a) for thunderstorms observed by TROPOMI over the U.S. ($0.41 \pm 0.10$) and are similar as the averaged $AMF_{LNOx}$ value in thunderstorms (0.46) reported by Beirle et al. (2009) over the Pacific.

Background-$NO_x$ values as the $30^{th}$ and the $10^{th}$ percentile of $V_{tropNOx}$ over non-flashing pixels with deep convection
($V_{tropbck}$) are shown in column 8. As in the case of $V_{tropNOx}$, we receive lower values of $V_{tropbck}$ than for the TROP-KNMI research product. There are even some negative values, suggesting that the average stratospheric column exceeding the local vertical column (eq. (3)) or the tropospheric background exceeding the signal (eq. (2)). $V_{tropbck}$ values show a large variability, although the mean values are of the same order as the background estimated from CARIBIC measurements (0.75 petamolec $cm^{-2}$) and from TROPOMI measurements over convective systems with low lightning activity (1.06 petamolec $cm^{-2}$ for the
TROP-KNMI product and 0.37 molec $cm^{-2}$ for the TROP-DLR research product), as detailed in Section 2.6.

The $LNO_x$ PE for each case using ENGLN and EUCLID lightning data are shown in column 9 and 10 of Tables 1 and 2, respectively. We have used the standard deviation over all cases in order to estimate the error of the mean PE. We can see a factor of $\sim$2 difference between the $LNO_x$ PE using different backgrounds for most of the cases, indicating that the method to estimate the background introduces a significant uncertainty of the results. Using the TROP-KNMI research product, we obtain
lower $LNO_x$ PE for ENGLN than for EUCLID ($47 \pm 33$ mol $NO_x$ per flash vs $69 \pm 34$ mol $NO_x$ per flash). On the contrary, we obtain slightly higher $LNO_x$ PE for ENGLN than for EUCLID when using the TROP-DLR product ($58 \pm 33$ mol $NO_x$ per flash vs $51 \pm 25$ mol $NO_x$ per flash). The mean $LNO_x$ PE values averaged over ENGLN and EUCLID for the TROP-KNMI and the TROP-DLR products are 58 and 54.5 mol $NO_x$ per flash, respectively. The $LNO_x$ PE value using the TROP-KNMI product is then higher than the value using the TROP-DLR product. We suggest that this slight difference is caused by the
higher stratospheric VCD $NO_2$ value in the TROP-DLR product.

The standard deviations of the $LNO_x$ PE derived from the TROP-DLR and the TROP-KNMI products are rather similar, suggesting that the variability in the concentration of $NO_2$ provided by the TROP-DLR $NO_2$ product is similar to the variability provided by the TROP-KNMI product.

The average number of pixels with deep convection and satisfying the quality criterion using the TROP-KNMI product is
370, while it is 758 for the TROP-DLR product. This difference is a consequence of the cut-off employed for both the retrieved cloud fraction and OCP. The cloud fraction over the studied cases is about 30% larger for the TROP-DLR product than for the TROP-KNMI product, while the OCP is about 10% lower for the TROP-DLR product than for the TROP-KNMI product, leading to more pixels with deep convection in the case of TROP-DLR product than in the case of TROP-KNMI product. We have found that using 650 hPa as OCP threshold for the TROP-KNMI product instead of 523 hPa produces a similar
total number of pixels with deep convection and satisfying the quality criterium using the TROP-KNMI and the TROP-DLR





products. This change in the OCP threshold for the TROP-KNMI product produces a change of only +14% in the $LNO_x$ PE estimates, as more pixels with low convection would be included in the estimation of the background-$NO_x$.

### 3.2.2 $LNO_x$ PE estimates using fixed background-$NO_x$ values

Let us now estimate the average $LNO_x$ PE over all cases using the *background-$NO_x$ based on days with low lightning activity*
as calculated in Section 2.6. Instead of using the $V_{tropbck}$ values of Tables 1 and 2, we use 1.06 petamolec cm$^{-2}$, and 0.37 peta-molec cm$^{-2}$ for estimations of the $LNO_x$ PE based on the TROP-KNMI and the TROP-DLR research products, respectively. We obtain $86 \pm 63$ mol $NO_x$ per flash by using the TROP-KNMI product with ENGLN lightning data, $160 \pm 102$ mol $NO_x$ per flash by using the TROP-KNMI product with EUCLID lightning data. These values are larger than the mean $LNO_x$ PE using non-flashing pixels ($47 \pm 33$ and $69 \pm 34$ mol $NO_x$ per flash).

By using the background-$NO_x$ based on days with low lightning activity, we calculate $44 \pm 61$ mol $NO_x$ per flash by using the TROP-DLR product with ENGLN lightning data and $53 \pm 59$ mol $NO_x$ per flash using the TROP-DLR product with EUCLID lightning data. The $LNO_x$ PE estimates based on the TROP-DLR product for the two cases of 7 May and 12 May are negative when using the background-$NO_x$ based on days with low lightning activity, causing lower values of $LNO_x$ PE and larger standard deviations than using the TROP-KNMI product. These values are in agreement with the mean $LNO_x$ PE using
non-flashing pixels ($58 \pm 33$ and $51 \pm 25$ mol $NO_x$ per flash).

We calculate the average $LNO_x$ PE over all cases by using the *background-$NO_x$ estimated from CARIBIC measurements* (0.75 petamolec cm$^{-2}$), as described in Section 2.6. We obtain $96 \pm 67$ mol $NO_x$ per flash using the TROP-KNMI product with ENGLN lightning data, $176 \pm 108$ mol $NO_x$ per flash using the TROP-KNMI product with EUCLID lightning data. These values are larger than the mean $LNO_x$ PE using non-flashing pixels ($47 \pm 33$ and $69 \pm 34$ mol $NO_x$ per flash). Finally,
we calculate $17 \pm 48$ mol $NO_x$ per flash by using the TROP-DLR product with ENGLN lightning data and $34 \pm 74$ mol $NO_x$ per flash using the TROP-DLR product with EUCLID lightning data. Again, the standard deviation of the TROP-DLR $LNO_x$ PE using a fixed value as background-$NO_x$ mixing ratio is lower than in the previous cases, as a consequence of low VCD $NO_x$ of the cases 12 May and 7 May. The $LNO_x$ PE estimates using the TROP-DLR product are negative because the tropospheric VCD of $NO_x$ is lower than the CARIBIC-based estimated background-$NO_x$ (fourth column in Table 2). The
obtained TROP-DLR values are lower than the mean $LNO_x$ PE using non-flashing pixels ($58 \pm 33$ and $51 \pm 25$ mol $NO_x$ per flash).

Given that the standard deviation of the received $LNO_x$ PE estimates by using fixed values of the background-$NO_x$ are larger than the means for the TROP-DLR product, we conclude that using fixed values for the background is not adequate in this case-based study. This is a consequence of the observed large variability of the tropospheric VCD of $NO_x$ for each studied
thunderstorms. Fixed background values could be useful to estimate the mean $LNO_x$ PE over a number of case studies but less useful to individual case studies.





### 3.3 Sensitivity analysis and uncertainties

In this section we discuss the most important uncertainties in the estimation of $LNO_x$ PE presented in section 3.2.1. We calculate the uncertainty associated with each parameter by comparing the maximum and the minimum received $LNO_x$ PE

values to the mean of the value for the possible choices of that parameter.

Let us begin by discussing the contribution of the employed lightning data to the uncertainty of the $LNO_x$ PE estimates. The mean $LNO_x$ PE of both TROPOMI products (KNMI and DLR) by using ENGLN lightning data is 52.5 mol $NO_x$ per flash, while it is 60 mol $NO_x$ per flash using EUCLID lightning data. Therefore, the *uncertainty introduced by different lightning data sets* is 7%. We have calculated the T-test for the means of the $LNO_x$ PE estimates when using ENGLN and EUCLID

lightning data, obtaining a p-value of 0.43. Therefore, we conclude that differences in $LNO_x$ PE using ENTLN and EUCLID are not statistically significant.

The $LNO_x$ PE estimates by *using different TROPOMI products (KNMI versus DLR)* are not similar, as obtained in section 3.2.1. There is a 23% difference between the $LNO_x$ PE estimates using both TROPOMI products and ENGLN lightning data, and a 35% difference when using EUCLID lightning data. The difference is reduced when using only ENGLN lightning

data, whose DE is higher than for EUCLID. The total uncertainty introduced by the choice of the TROPOMI product based on the means $LNO_x$ PE per flash between ENGLN and EUCLID lightning data is only 3%. We obtain a p-value of 0.44 by calculating the T-test for the means of the $LNO_x$ PE estimates when using TROP-KNMI and TROP-DLR, indicating that differences in $LNO_x$ PE using different TROPOMI products are not statistically significant.

As shown in Tables 1 and 2, the estimation of the *background-$NO_x$* as the $30^{th}$ or as the $10^{th}$ percentile of $V_{tropNOx}$ over

non-flashing pixels with deep convection can significantly influence the $LNO_x$ PE estimates. The average $LNO_x$ PE between both TROPOMI products using the $30^{th}$ percentile of $V_{tropNOx}$ is 42 mol $NO_x$ per flash, while it is 70 mol $NO_x$ per flash using the $10^{th}$ percentile of $V_{tropNOx}$. Therefore, the choice of the background-$NO_x$ method contributes to the uncertainty of 29%. The p-value obtained by calculating the T-test for the means of the $LNO_x$ PE estimates by using the $30^{th}$ or the $10^{th}$ percentile of $V_{tropNOx}$ over non-flashing pixels with deep convection as background-$NO_x$ is lower than 0.05, which indicates

that differences in $LNO_x$ PE using different methods to estimate the background-$NO_x$ products are statistically significant.

The *DE of the used LLS* can also contribute to the uncertainty of the $LNO_x$ PE estimates. As explained in section 2.2, we obtain a DE for ENGLN over the Pyrenees of $0.676 \pm 0.12$ (ranging between 0.556 and 0.769 ). The obtained mean $LNO_x$ PE using both TROPOMI products and a DE of 0.769 is 59 mol $NO_x$ per flash, while it is 43 mol $NO_x$ per flash when using a DE of 0.556. Therefore, the uncertainty of the DE of ENGLN contributes to a $LNO_x$ PE uncertainty of 17%. For EUCLID, we

obtain a DE of $0.27 \pm 0.12$. The obtained mean $LNO_x$ PE using EUCLID data corrected by a DE of 0.40 is 86 mol $NO_x$ per flash, while it is 33 mol $NO_x$ per flash when using a DE of 0.15. Therefore, the uncertainty of the DE of EUCLID contributes to a $LNO_x$ PE uncertainty of 62%. The contribution of the DE of EUCLID to the uncertainty is higher than the contribution of the DE of ENGLN because the DE of EUCLID is significantly lower than the DE of ENGLN.

The *lifetime of $NO_x$* in the near field of convection ($\tau$) is another parameter that can introduce uncertainty to the $LNO_x$ PE

estimates. We have used 3 h, but it can vary between 2 and 12 h (Nault et al., 2017; Allen et al., 2021a). We have performed





**Table 3.** Sources of differences in the mean LNO$_x$ PE estimates.

| Source of difference | Influence on the LNO$_x$ PE estimate |
| --- | --- |
| Lightning data set (ENGLN or EUCLID) | 7% |
| TROPOMI product (DLR or KNMI v2.1) | 3% |
| Background-NO$_x$ estimation (10% or 30% of non-flashing pixels) | 29% |
| Lightning detection system DE using ENGLN | 17% |
| Lightning detection system DE using EUCLID | 62% |
| Lifetime of NO$_x$ in the near field of convection ($\tau$) | 18% |
| Time window before the TROPOMI overpass | 29% |
| Other (lightning parameterization, scattering weights, deep convection definition) | 30% |

the LNO$_x$ PE calculations using the TROPOMI products and ENGLN lightning data and setting $\tau$ = 12 h as an upper limit keeping the time windows used at 5 h, obtaining a mean LNO$_x$ PE of 38 mol NO$_x$ per flash. Given that the LNO$_x$ PE with $\tau$ = 3 h is 52.5 mol NO$_x$ per flash, we estimate that $\tau$ contributes to the uncertainty of the LNO$_x$ PE by about 18%.

The *time window before the TROPOMI overpass*, that is used to count the total number of lightning flashes contributing to

fresh-produced LNO$_x$, can also be a source of uncertainty. We have calculated the LNO$_x$ PE estimates using a time window of 1 h instead of 5 h in order to get an estimation of the uncertainty introduced by the time window. We receive 88 mol NO$_x$ per flash as the mean value by using the TROP-KNMI and the TROP-DLR products and ENGLN lightning data. The LNO$_x$ PE estimations using the same TROPOMI products and lightning data with a time window of 5 h was 52.5 mol NO$_x$ per flash. According to our estimations, the time window contribution to the uncertainty of the LNO$_x$ PE is about 29%. We do not

perform calculations using a larger time window, because studying the transport of LNO$_x$ at longer time scales is out of the scope of this work.

The sources of differences in the LNO$_x$ PE estimation evaluated in this study are summarized in Table 3. As discussed in previous studies (e.g., Pickering et al., 2016; Allen et al., 2019; Lapierre et al., 2020; Zhang et al., 2020; Allen et al., 2021a), there are other possible sources of uncertainty, such as the calculation of the $AMF$ (LNO$_x$ profile type and lightning

parameterization and NO$_x$/NO$_2$ ratios in the simulations, scattering weights calculations) contributing to the uncertainty of about 30% or the method to select the OCP to be used for the definition of deep convection, contributing to the uncertainty of about 10%, or other systematic errors in the retrieval algorithms of TROPOMI. However, estimates of the influence of these parameters for the uncertainty of LNO$_x$ PE on the particular area of the Pyrenees is out of the scope of this paper, as we do not expect them to be dependent on the studied area.

We can estimate the overall LNO$_x$ PE uncertainty by summing the uncertainties in PE collected in Table 3. We obtain an overall LNO$_x$ PE uncertainty of 57% using ENGLN lightning data and 83% using EUCLID lightning data.

## 4 Discussion

Previous studies have used OMI NO$_2$ measurements to estimate the LNO$_x$ PE over different regions, as shown in Table 4. Pickering et al. (2016) reported a LNO$_x$ PE of 80 $\pm$ 45 mol per flash over the Gulf of Mexico. Bucsela et al. (2019) system-





**Table 4.** Some recent LNO$_x$ PE estimates.

| Area | Instrument | LNO$_x$ PE estimate (mol per flash) | Reference |
|---|---|---|---|
| Gulf of Mexico | OMI | $80 \pm 45$ | Pickering et al. (2016) |
| Mid-latitudes | OMI | $180 \pm 100$ mol | Bucsela et al. (2019) |
| Tropics | OMI | $170 \pm 100$ | Allen et al. (2019) |
| USA | OMI | $\sim 24$ mol | Lapierre et al. (2020) |
| USA | OMI | $90 \pm 50$ | Zhang et al. (2020) |
| USA | TROPOMI | $120 \pm 50$ | Allen et al. (2021a) |
| Pyrenees and Ebro Valley | TROPOMI | $58 \pm 44$ | This work |

atically estimated the LNO$_x$ PE over mid-latitudes, obtaining an average LNO$_x$ PE of $180 \pm 100$ mol per flash. Interestingly, Bucsela et al. (2019) (see Table 1) found a lower LNO$_x$ PE in Europe ($150 \pm 90$ mol per flash). Allen et al. (2019) reported a mean LNO$_x$ PE over the tropics of $170 \pm 100$ mol per flash. Lapierre et al. (2020) reported a LNO$_x$ PE over the USA of $\sim 24$ mol per flash (estimated from mol per stroke calculations), while Zhang et al. (2020) reported $90 \pm 50$ mol per flash over the USA. Recently, Allen et al. (2021a) have estimated the LNO$_x$ PE in 29 thunderstorms over the USA by using new TROPOMI

NO$_2$ data, finding a LNO$_x$ PE of $120 \pm 50$ mol per flash based on the use of ENGLN lightning data. We have calculated the T-test for the means of the LNO$_x$ PE estimates when using ENGLN lightning data together with the TROP-KNMI product and the LNO$_x$ PE estimates provided by Allen et al. (2021a) when using ENGLN lightning data, obtaining a p-value lower than 0.05. Therefore, we conclude that differences in LNO$_x$ PE between the Pyrenees and the U.S. are statistically significant.

We have used the LNO$_x$ PE algorithm employed by Pickering et al. (2016); Bucsela et al. (2019); Allen et al. (2019) and

Allen et al. (2021a) to provide new LNO$_x$ PE estimate based on TROPOMI NO$_2$ measurements over the Pyrenees. We obtain $47 \pm 33$ ($69 \pm 34$) mol NO$_x$ per flash using the TROP-KNMI research product and ENGLN (EUCLID) lightning data and $58 \pm 33$ ($51 \pm 25$ mol NO$_x$) mol NO$_x$ per flash using TROP-DLR product and ENGLN (EUCLID) lightning data. Our mean LNO$_x$ PE estimates are slightly lower than the LNO$_x$ PE reported by Pickering et al. (e.g., 2016); Allen et al. (e.g., 2019); Zhang et al. (e.g., 2020); Allen et al. (e.g., 2021a) and a factor of $\sim 2$ higher as determined by Lapierre et al. (2020).

Let us now compare our results with TROPOMI-based estimates by Allen et al. (2021a) over the USA using ENGLN lightning data ($120 \pm 50$ mol). We obtain lower LNO$_x$ PE estimates, which is in agreement with Bucsela et al. (2019), who reported a lower LNO$_x$ PE over Europe than over the USA. We estimate a mean tropospheric VCD of NO$_x$ of 3.5 petamolec cm$^{-2}$ from the TROP-KNMI product. Allen et al. (2021a) reported a slightly higher mean VCD of NO$_x$ of 4.4 petamolec cm$^{-2}$ from the TROP-KNMI product. The Pyrenees are a low contaminated area, which explains that the tropospheric VCD

of NO$_x$ is lower than for the 29 cases studied by Allen et al. (2021a) over the USA. We have also found comparable influence of the background-NO$_x$ on the uncertainty of our results than Allen et al. (2021a), (29% vs 22.5%). The explanation of this difference can be that Allen et al. (2021a) analyzed 29 cases, while in this study we have analyzed only 8 cases.

The obtained LNO$_x$ PE are significantly influenced by the TROPOMI (KNMI and DLR) and the lightning (ENGLN and EUCLID) data sets. The difference between the LNO$_x$ PE calculated by using the TROP-KNMI and the TROP-DLR products

together with the ENGLN lightning data is 3%. There is a factor of 3.5 difference in the estimated median tropospheric VCD of NO$_x$ using the TROP-KNMI product (3.5 petamolec cm$^{-2}$) and the TROP-DLR product (0.96 petamolec cm$^{-2}$), while the



differences in the provided mean stratospheric VCD of $NO_2$ over pixels with deep convection is 14% (7.3 and 8.3 petamolec $cm^{-2}$ for the TROP-KNMI and the TROP-DLR products, respectively). The background-$NO_x$ is estimated from non-flashing pixels, leading to a similar $V_{tropLNOx}$ and $LNO_x$ PE values. However, using a fixed value for the background-$NO_x$ produces

significantly lower $LNO_x$ PE for the TROP-DLR product than for the TROP-KNMI product, as a consequence of the lower tropospheric VCD of $NO_x$ obtained from the TROP-DLR product.

Despite significant differences in the DE of ENGLN and EUCLID in the studied area, we have not found significant differences in the mean estimation of the $LNO_x$ PE using lightning data from both networks after correction with the DE. The $LNO_x$ PE estimates using the TROP-DLR product together with ENGLN and EUCLID lightning data are nearly similar (58

$\pm$ 33 mol $NO_x$ per flash and 51 $\pm$ 25 mol $NO_x$, respectively). However, we have found that the $LNO_x$ PE obtained using the TROP-KNMI product are different for ENGLN (47 $\pm$ 33 mol per flash) and EUCLID data (69 $\pm$ 34 mol per flash). We have found that the received $LNO_x$ PE using ENGLN ranges between 39 and 59 mol $NO_x$ per flash after correction by the DE 0.676 $\pm$ 0.12, while the calculated $LNO_x$ PE using EUCLID ranges between 33 and 86 mol $NO_x$ per flash after correction by the DE 0.27 $\pm$ 0.12. Therefore, we conclude that the higher DE of ENGLN provides more precise $LNO_x$ PE than EUCLID in the

studied area.

## 5 Conclusions

We have estimated the $LNO_x$ PE over the Pyrenees, a European region with high lightning activity and relatively low concentration of background-$NO_x$. We have used two lightning data sets (ENGLN and EUCLID) and two TROPOMI $NO_2$ and cloud products (DLR and KNMI v2.1) in this study. The main conclusions of this work are as follows:

1. We obtain 47 $\pm$ 33 mol $NO_x$ per flash using the TROP-KNMI research product and ENGLN lightning data, 69 $\pm$ 34 mol $NO_x$ per flash using TROP-KNMI research product and EUCLID lightning data, 58 $\pm$ 33 mol $NO_x$ per flash using the TROP-DLR product and ENGLN lightning data and 51 $\pm$ 25 mol $NO_x$ per flash by using TROP-DLR product and EUCLID lightning data. Overall, the obtained $LNO_x$ PE ranges between 14 and 103 mol $NO_x$ per flash. These estimates are lower than the globally averaged $LNO_x$ PE (250 $\pm$ 150 mol per flash) estimated by Schumann and Huntrieser (2007) 495 and the $LNO_x$ PE estimates from the TROPOMI measurements and ENGLN lightning data in the USA by Allen et al. (2021a) (120 $\pm$ 50 mol per flash).

2. We have used different methods to estimate the background-$NO_x$, i.e., the background-$NO_x$ from non-flashing pixels and from measurements over days with low lightning activity. When using ENGLN lightning data, we have found that the most important source of uncertainty for $LNO_x$ PE is the estimation of the background-$NO_x$ (about 29%), similar 500 as by the time window prior to the TROPOMI overpass time used to collect the lightning data (about 29%). The overall uncertainty when using ENGLN lightning data is 57%. When using EUCLID lightning data, the most important source of uncertainty is the DE of EUCLID (about 62%), while the overall uncertainty when using EUCLID lightning data is 83%.





3. The estimated median tropospheric VCD of $NO_x$ in convective systems after subtraction of the stratospheric $NO_2$ contribution

is a factor of 3.5 lower for the TROP-DLR product than for the TROP-KNMI product as a consequence of larger stratospheric VCD of $NO_2$ in the TROP-DLR product over pixels with deep convection.

4. The mean $LNO_x$ PE obtained from ENGLN and EUCLID lightning data are not considerably different in comparison to differences obtained by using different methods to estimate the background-$NO_x$.

This paper reports on partly new and partly established methods to estimate $LNO_x$ PE. It confirms that the uncertainty in the

calculation of $LNO_x$ PE is still high, even when using high resolution measurements from TROPOMI. It also suggests that the $LNO_x$ PE vary substantially between different regions, as suggested by a comparison between our results and recent OMI- and TROPOMI-based $LNO_x$ PE over the USA (Lapierre et al., 2020; Allen et al., 2021a). This study also shows that differences in $LNO_x$ PE estimates can be caused by the different lightning systems.

The launch of the Meteosat Third Generation (MTG) geostationary satellites of the EUropean organization for the ex-

ploitation of METeorological SATellites (EUMETSAT) in 2022 will for the first time provide a continuous monitoring of the occurrence of lightning flashes from space in Europe and Africa through the instrument Lightning Imager (LI) from 2023 onwards (Stuhlmann et al., 2005). Lightning data from the MTG-LI can contribute to improve $LNO_x$ estimates over the studied region, Europe and Africa. In fact, lightning data from the geostationary GLM has already contributed to new $LNO_x$ PE estimations over America (Allen et al., 2021b, a). High temporal and spatial resolution observations from the Geostationary

Environment Monitoring Spectrometer (GEMS) and the future $NO_2$ retrieving instruments on-board geostationary satellites, such as the SENTINEL-4 GEO in 2023 (Courrèges-Lacoste et al., 2017), the Tropospheric Emissions: Monitoring of Pollution (TEMPO) (Zoogman et al., 2017) in 2022 will also contribute to provide more data to estimate the $LNO_x$ PE over Asia, North America, Europe and Africa.

*Data availability.* All data used in this paper are directly available after a request is made to authors F. J. P. I (FranciscoJavier.Perez-

Invernon@dlr.de) or H. H. (Heidi.Huntrieser@dlr.de). The official TROPOMI data are available via ESA's public data hub (https://s5phub. copernicus.eu/). The ERA5 meteorological data are freely accessible through Copernicus Climate Change Service (C3S) (2017): ERA5: Fifth generation of ECMWF atmospheric reanalyses of the global climate Copernicus Climate Change Service Climate Data Store (CDS) (https://cds.climate.copernicus.eu/cdsapp). ENGLN and EUCLID data were obtained freely by request from Earth Networks (https://www. earthnetworks.com) and AEMET (http://www.aemet.es/es/datos_abiertos), respectively. ISS-LIS data can be freely downloaded from https://

ghrc.nsstc.nasa.gov/lightning/data/data_lis_iss.html. IAGOS-CARIBIC data can be freely downloaded from https://www.iagos.org/iagos-data/.

# A    Appendix A: Acronym list of physical quantities

A: Area of deep convection

    AMF: Ais mass factor





$AMF_{LNOx}$: Ais mass factor used to convert tropospheric slant columnsdensity of $NO_2$ to vertical column density of $LNO_x$

$AMF_{strat}$: Stratosferic air mass factor from TROPOMI product

DE: Detection efficiency

F: Flashes contributing to $LNO_x$ column

$N_A$: Avogadro's number

OCP: Optical centroid pressure

PE: Production efficiency

SCD: Slant column density

$S_{NO2}$: Slant column density of $NO_2$

$t_i$: Time of individual flash

$\tau$: Chemical lifetime of $LNO_x$

VCD: Vertical column density

$V_{tropbkgn}$: Vertical column of tropospheric $NO_x$ due to non-recent lightning

$V_{tropLNOx}$: Vertical column of tropospheric $NO_x$ due to recent lightning

$V_{tropNOx}$: Vertical column of tropospheric $NO_x$

*Author contributions.* F.J.P.I.: Conceptualization, methodology, validation, formal analysis, investigation, data curation, writing—original

draft. H.H.: Conceptualization, methodology, validation, formal analysis, supervision, investigation, writing—review and editing. T. E.: Validation, data curation. D. L., P. V. and S.L: Validation, data curation, preparation of the TROP-DLR product. D. A., K. P. and E. B: Methodology, validation, formal analysis. P. J.: Validation, supervision of EMAC simulations. J. v. G and H. E: Validation, data curation, preparation of the TROP-KNMI product. F.J.G.V., S. S. and J. L.: Data curation, validation, preparation of the ENGLN lightning data.

*Competing interests.* Authors declare no competing interests.

*Acknowledgements.* The authors would like to thank DLR and KNMI for providing TROPOMI research $NO_2$ and cloud data, Earth Networks for providing ENTGN lightning data, Spanish State Meteorological Agency (AEMET) for providing EUCLID lightning data, NASA for providing ISS-LIS lightning data, ECMWF for providing the data of ERA5 forecasting models and IAGOS Research Infrastructure for providing NO data. The EMAC simulations have been performed at the German Climate Computing Centre (DKRZ) through support from the Bundesministerium für Bildung und Forschung (BMBF). DKRZ and its scientific steering committee are gratefully acknowledged for

providing the HPC and data archiving resources.Authors would also like to thank Volker Grewe (Deutsches Zentrum für Luft-und Raumfahrt, DLR) for providing valuable comments on this manuscript.

FJPI acknowledges the sponsorship provided by the Federal Ministry for Education and Research of Germany through the Alexander von Humboldt Foundation. Additionally, this work was supported by the Spanish Ministry of Science and Innovation, under projects PID2019-



109269RB-C43 and FEDER program. SS acknowledges a PhD research contract through the project PID2019-109269RB-C43. FJGV and SS

acknowledge financial support from the State Agency for Research of the Spanish MCIU through the 'Center of Excellence Severo Ochoa' award for the Instituto de Astrofísica de Andaluca (SEV-2017-0709).



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



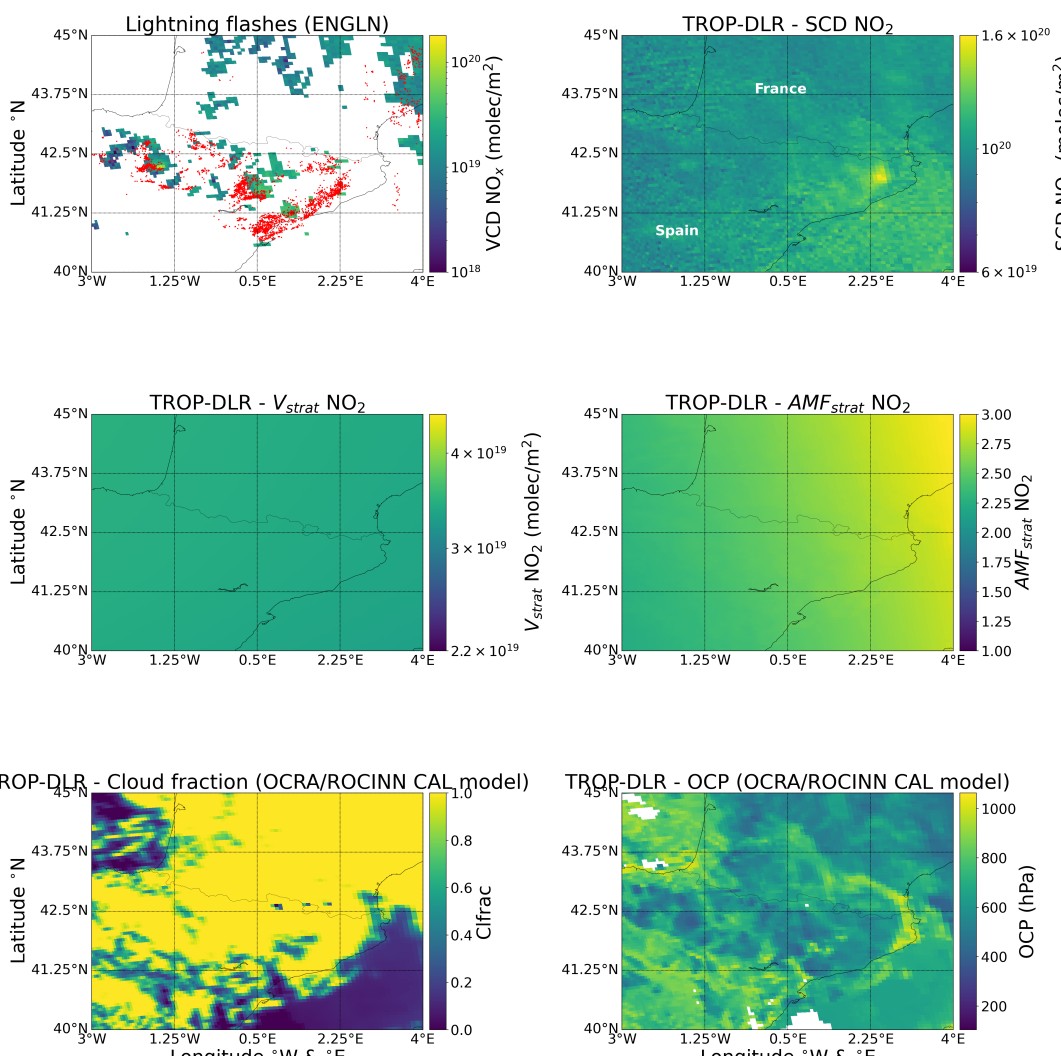

**Figure 6.** TROP-DLR product and ENGLN lightning data for the case 29 April 2018. The upper left panel shows the positions of lightning flashes (red dots) reported by ENGLN during the 5 h period before the TROPOMI overpass and the calculated VCD $NO_x$. The upper right panel shows the SCD of $NO_2$, center left and right panels show the stratospheric VCD and AMF of $NO_2$. The lower left and right panels show the cloud fraction and the OCP, respectively.



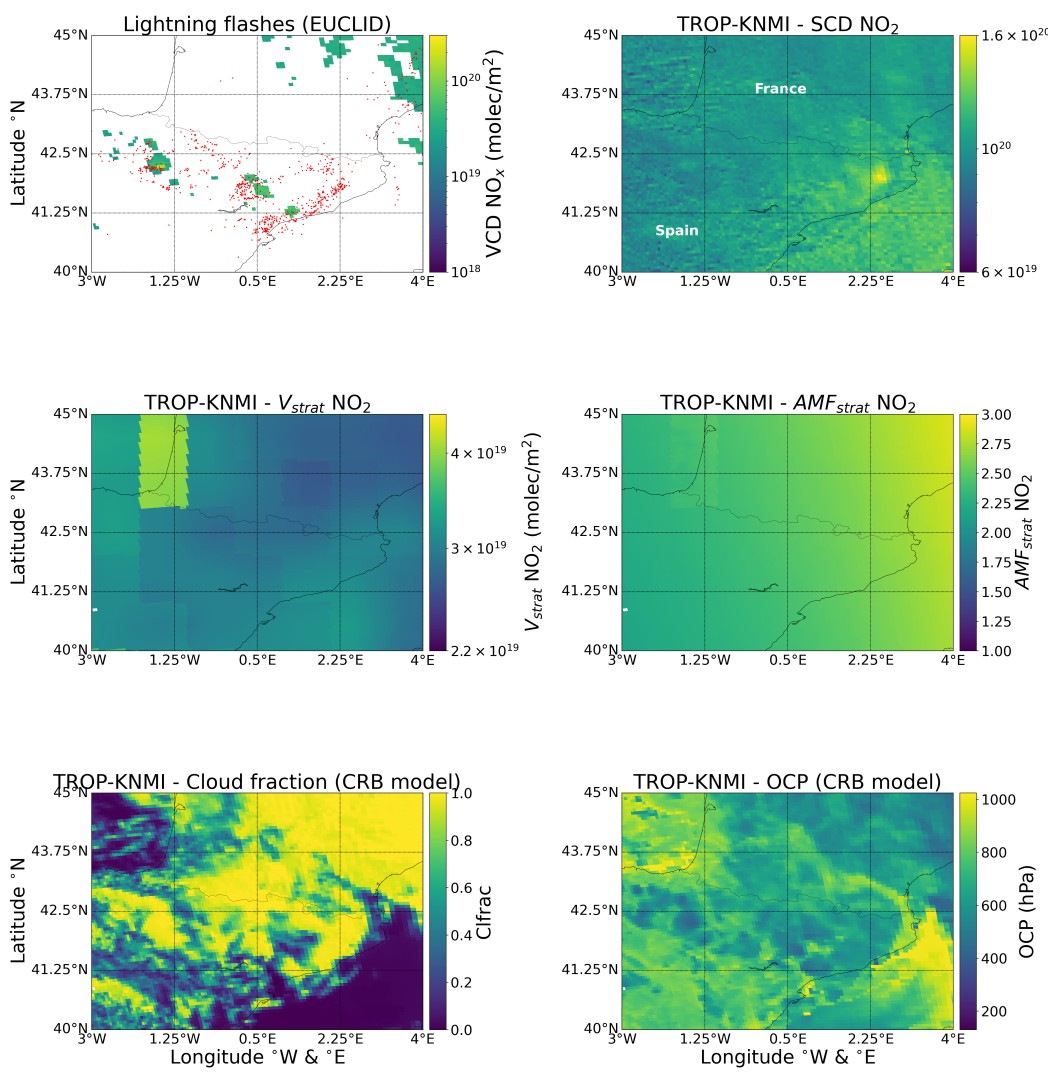

**Figure 7.** TROP-KNMI product and EUCLID lightning data for the case 29 April 2018. The upper left panel shows the positions of lightning flashes (red dots) reported by EUCLID during the 5 h period before the TROPOMI overpass and the calculated VCD $NO_x$. The upper right panel shows the SCD of $NO_2$, center left and right panels show the stratospheric VCD and AMF of $NO_2$. The lower left and right panels show the cloud fraction and the OCP, respectively.

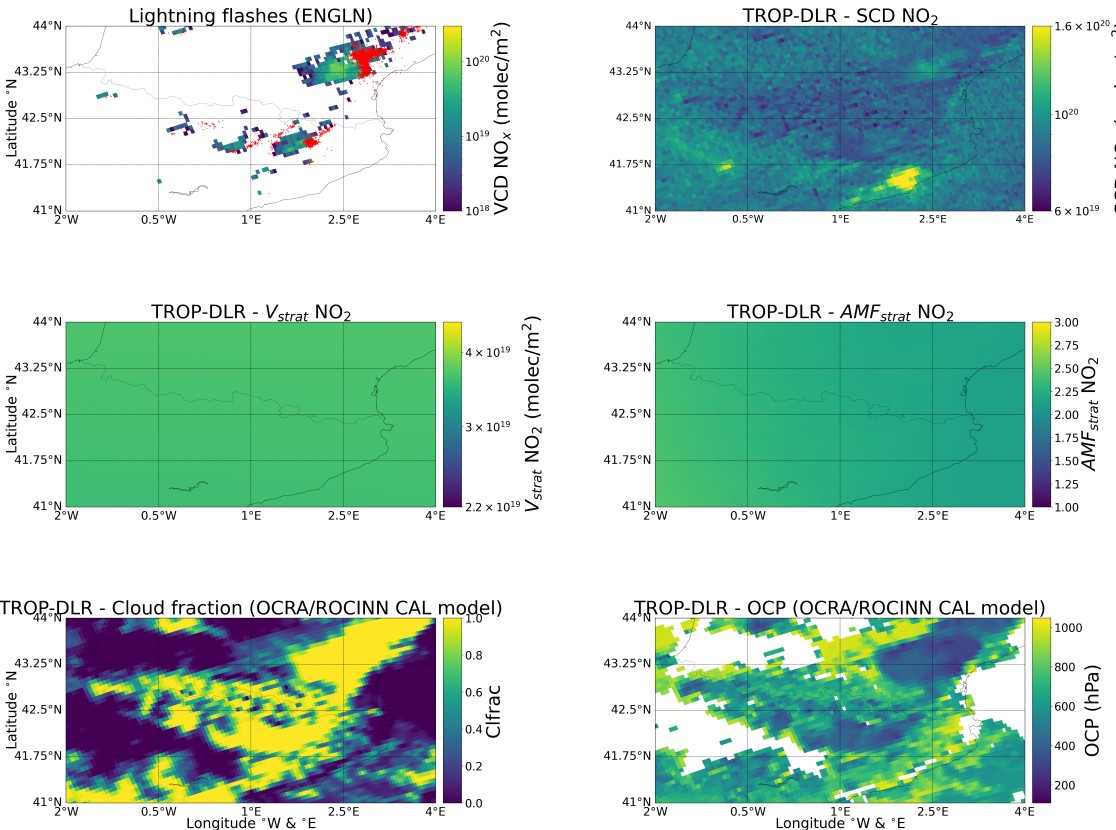

**Figure 8.** TROP-DLR product and ENGLN lightning data for the case 7 May 2018. The upper left panel shows the positions of lightning flashes (red dots) reported by ENGLN during the 5 h period before the TROPOMI overpass and the calculated VCD $NO_x$. The upper right panel shows the SCD of $NO_2$, center left and right panels show the stratospheric VCD and AMF of $NO_2$. The lower left and right panels show the cloud fraction and the OCP, respectively.

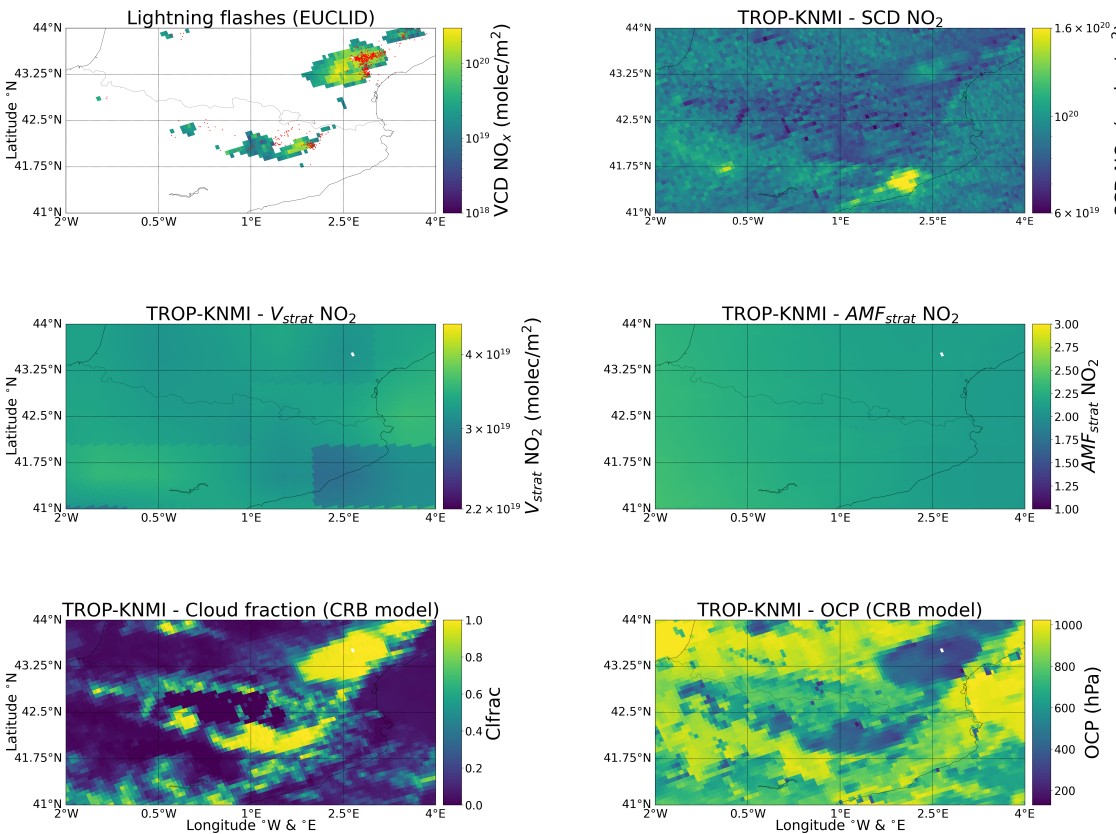

**Figure 9.** TROP-KNMI product and EUCLID lightning data for the case 7 May 2018. The upper left panel shows the positions of lightning flashes reported by EUCLID (red dots) reported by ENGLN during the 5 h period before the TROPOMI overpass and the calculated VCD $NO_x$. The upper right panel shows the SCD of $NO_2$, center left and right panels show the stratospheric VCD and AMF of $NO_2$,. The lower left and right panels show the cloud fraction and the OCP, respectively.

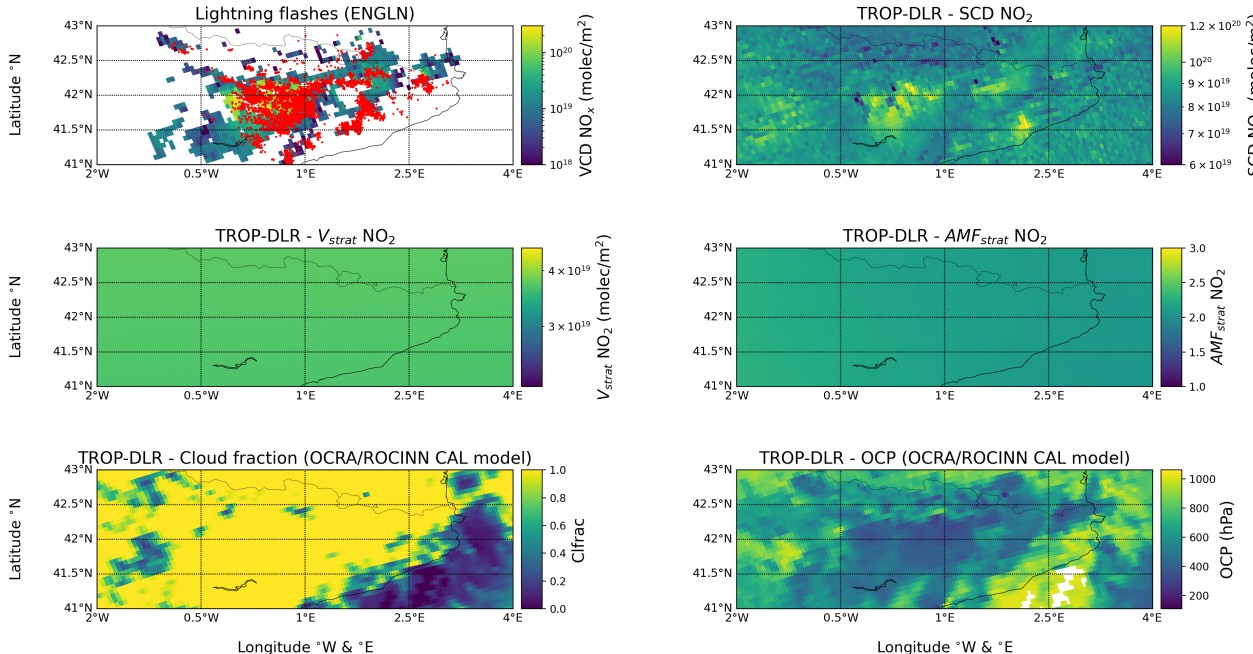

**Figure 10.** TROP-DLR product and ENGLN lightning data for the case 28 May 2018. The upper left panel shows the positions of lightning flashes (red dots) reported by ENGLN during the 5 h period before the TROPOMI overpass and the calculated VCD $NO_x$. The upper right panel shows the SCD of $NO_2$, center left and right panels show the stratospheric VCD and AMF of $NO_2$. The lower left and right panels show the cloud fraction and the OCP, respectively.

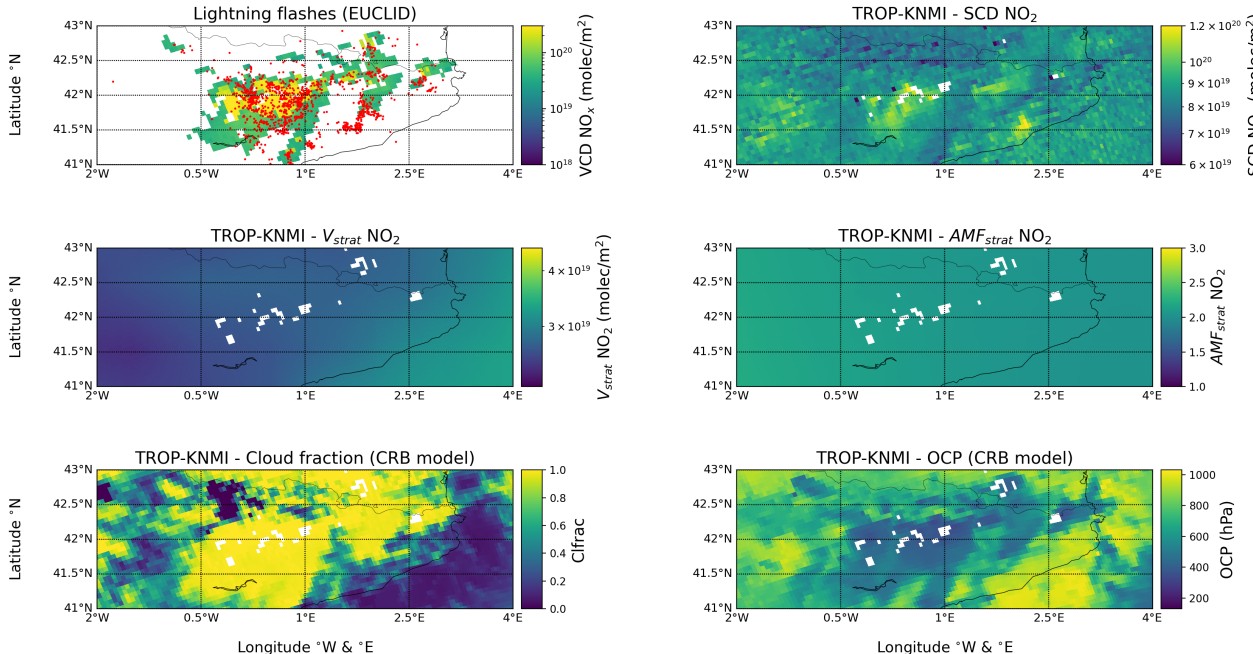

**Figure 11.** TROP-KNMI product and EUCLID lightning data for the case 28 May 2018. The upper left panel shows the positions of lightning flashes (red dots) reported by EUCLID during the 5 h period before the TROPOMI overpass and the calculated VCD $NO_x$. The upper right panel shows the SCD of $NO_2$, center left and right panels show the stratospheric VCD and AMF of $NO_2$. The lower left and right panels show the cloud fraction and the OCP, respectively.



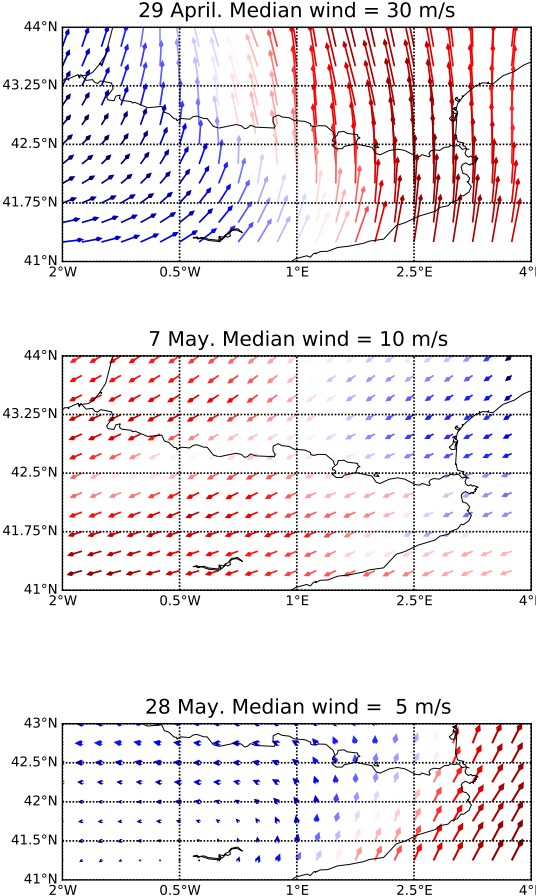

**Figure 12.** Horizontal wind velocity and direction averaged between 200 hPa and 500 hPa pressure levels for the studied cases on 29 April,
7 May and 28 May, 2018. The horizontal winds are extracted from ERA5-reanalysis data.