# Peer review of "Quantification of lightning-produced $NO_x$ over the Pyrenees and the Ebro Valley by using different TROPOMI-NO2 and cloud research products"

_Atmospheric Measurement Techniques, 2021_

## Author Response (AR1)

**Comments from the quick report**

The manuscript is interesting, but there are quite some inclarities that should be resolved:

We thank the reviewer for these encouraging comments and for the time spent for the revision.

* One of the main conclusions of this paper is that the LNOx PE estimated from DLR's TROPOMI product is higher than from KNMI's NO2 product. Yet the tropospheric NO2 enhancements in the DLR product are a factor 3-4 lower than in the KNMI product, because of the larger correction for stratospheric NO2 by DLR (via STREAM) than by KNMI (via data assimilation in TM5-MP). How can much smaller NO2 enhancements still lead to higher lightning production efficiencies, when the lightning flash estimates used are identical between the two products?

We thank the reviewer for raising this question. As the reviewer points out, the tropospheric NO2 enhancements of the DLR product are by a factor of 3-4 lower than in the KNMI product. The LNOx using ENGLN from TROP-DLR is higher that from TROP-KNMI when using ENGLN lightning data. However, we found the opposite when using EUCLID data. The main reason of this apparent disagreement is the total number of pixels with deep convection and satisfying the quality criterion when using the two different TROPOMI products and the pixels that are labeled as background when using two different lightning data. As we stated in line 350:

*"The average number of pixels with deep convection and satisfying the quality criterion using the TROP-KNMI product is 370, while it is 758 for the TROP-DLR product."*

Therefore, the estimated LNOx when using different TROPOMI products and lightning data is not only a matter of the absolute tropospheric NO2 from each TROPOMI product. The estimated LNOx is also highly influenced by the total number of pixels that satisfy the deep convection and quality criteria, which are different from each product. The pixels that satisfy the criteria are used to estimate the background. In addition, the pixels that are labeled as background are different when using ENGLN or EUCLID. The estimated LNOx is a combination of all the mentioned factors.

This can be clearer seen when we use a fixed value for the background-NOx estimation (line 373). In this case, the absolute value of the tropospheric NO2 determines which LNOx PE estimate is higher (KNMI or DLR). When we use the background-NOx estimated from CARIBIC measurements ($0.75 \ 10^{19}$ molec m$^{-2}$) we obtain a higher LNOx PE for TROP-KNMI than for TROP-DLR (96 +- 67 mol-NOx per flash versus 17 +- 48 mol-NOx per flash).

* The paper could improve in clarity and become more convincing if the observed NO2 enhancements would be visibly related to the location of the flashes via the wind vector. Now readers have to guess that such a relationship between the location of the NO2 enhancement and that of the flashes might exist, from staring at another figure showing the wind vector. I encourage the authors to make a more quantitative effort in linking up flashes and NO2, for example through an approach such as exploited by Georgoulias et al. [2020], who could attribute plumes of NO2 to individual ship tracks.

The purpose of using the wind in this study is to introduce a conservative method to exclude cells that could have been influenced by LNOx in the background-NOx estimation. The NO2 produced by ships is emitted near the surface. However, the NOx produced by lightning follows a complex vertical emission profile. A detailed calculation on the advection of LNOx would require a mesoscale model and is out of the scope of this paper. We show the winds averaged across some vertical levels in Fig. 12 to give the reader an insight on how we exclude cells in the background-NOx estimation.

\* I find the concept of an LNOx air mass factor difficult. I expect the authors will point to the work by Allen et al., but this is not of a review of that work, so I bring it up here. TROPOMI detects NO2 and not NOx column densities along the average (slant) photon path between the Sun and the satellite. To convert these slant columns into interpretable vertical columns, an NO2 AMF is required. So it is misleading to use a NOx AMF. I think the authors should present NO2 AMF values, which have a clear meaning and apply to the actual satellite retrievals. That the authors ultimately convert NO2 to NOx since flashes produce NOx rather than just NO2, can still proceed via subsequent correction factors, but not through a confusing inclusion in the AMF. Furthermore it is unclear how the AMFs are calculated, and why the KNMI and DLR AMFs are any different, presumably because of different cloud input parameters? This needs to be clarified.

As the reviewer points out, lightning produces NOx, while TROPOMI detects NO2 slant column density. However, using an AMF NO2 instead of a AMF NOx is not appropriate in this case. The meaning of the AMF NOx and the convenience of using it instead of AMF NO2 was explained by Beirle et al. (2009) before the studies of Allen et al. (2019 & 2021). As Beirle et al. (2009) pointed out:

"Please note that a two-step conversion (first from NO2 SCDs into NO2 VCDs using an overall AMF, and then from NO2 VCDs into NOx VCDs using a mean NO2/NOx ratio) is not appropriate, since both the box-AMFs and the NOx partitioning are height dependent, and they do not vary independently because both are particularly influenced by clouds."

This also answers the question about differences in AMF NOx for TROP-KNMI and TROP-DLR. The AMF NOx depends on the cloud input parameters, that are different for each of the employed TROPOMI products.

We have included these explanations in the manuscript.

\* The manuscript lacks a discussion of the differences between the KNMI and DLR cloud products. These differences presumably influence the AMF calculations, and are potentially important in figuring out a path to improve TROPOMI NO2 retrievals in the near future.

As can be seen in Fig. 2 (OCP values), in the fourth column of Tables 1 and 2, and in Figures 6 to 11, there are slight differences in the cloud products. We comment on these differences in lines 275 − 278:

"… the lower panels show that there are no significant differences between the cloud products, except for some pixels in which the TROP-DLR product estimates larger cloud fractions. The existence of more pixels with high cloud fractions in the TROP-DLR product, than in the TROP-KNMI product, can influence the total number of pixels labeled as cloud convective pixels."

We have added to the manuscript that differences between the TROP-KNMI and TROP-DLR products can also influence the estimation of the AMF LNOx parameter, as the pixels involved in the calculation are not the same.

Minor comments/suggestions
L80: was v2.1 KNMI NO2 data available before April-May 2021?

The v2.1 research product is not automatically produced for all the TROPOMI orbits. We produced it on a case-based demand to analyze particular thunderstorms. We have added this clarification to the manuscript.

L85: it would be appropriate to cite Williams et al. [2017], when referring to the TM5-MP version used for predicting NO2 profile shapes in TROPOMI NO2 retrievals.

Done.

Figure 2: have the cloud pressures shown in the figure been recorded for pixels with a cloud fraction >0.95 measured *after* flashes occurred? Over what time window have the flashes been accumulated?

The cloud pressures shown in Fig. 2 have been recorded for pixels where flashes occurred in the past 5 hours. We do not impose any cloud fraction here because the aim was to set a threshold in the OCP value, not in the cloud fraction. The cloud fraction is later introduced in the LNOx PE algorithm.

We have added the time window to the caption of Fig. 2.

Figure 3: why is the detection efficiency defined relative to the LIS?

The LIS instrument can detect the optical signature emitted by lightning from space. The DE of LIS is well known and quite constant around the globe (DE~ 0.6, [Blakeslee et al. (2020)]. In addition, using optical data to estimate the DE of a LSS is better than comparing lightning data acquired by a different LSS, as both systems could miss the same type of lightning (lightning producing low electromagnetic radio signals). These are the reasons by LIS is commonly used to estimate the DE of LSS (see for example Bitzer et al. (2016): https://doi.org/10.1002/2016GL071951). We have included the DE of ISS-LIS in the manuscript.

L136: should we understand that ENCL records 2/3rd of the flashes recorded by LIS?

No, we should not. As we explain in the manuscript, we use a Bayesian approach to estimate the DE of ENTLN.

L170: 0.45 ppb NO2 is on the high side compared to aircraft NO2 data in the high troposphere obtained in many campaigns (INTEX, ATOM, IAGOS, … ).

We use 0.45 ppb NO2 as an approximate value for areas with high convection. NO2 can be transported to the upper levels of the troposphere by convection, producing concentrations that are larger than typical values in the high troposphere.

Eq. (1): the choice of tau = 3 hrs seems to presuppose that in the 3 hours before TROPOMI overpass, the air mass remains at the location where the flashes occurred. In reality there will be wind, smearing out and transporting the signal from the flashes over a larger area. How are these effects accounted for in the background correction?

We agree with the reviewer. We have included the horizontal winds provided by ERA5 to account for this transport and exclude grid cells that could have been affected by LNOx in the estimation of the background-NOx. In addition, we use other values for tau in section 3.3 in order to quantify its influence on the results.

L237: please specify over how many hours prior to the TROPOMI overpass the number of flashes has been accumulated.

We have specified that we include flashes 3 hours prior to the TROPOMI overpass.

L271-273: how realistic are the tropopause jumps in the KNMI approach?

The tropopause jumps are predicted by the TM5-MP model. These jumps are common in the case of thunderstorms (see for example Pan et al. (2014): https://doi.org/10.1002/2014GL061921 ). We have added this to the manscript.

L273-275: differences in stratospheric NO2 columns between the STREAM and data assimilation methods have been reported on before (Boersma et al., AMT, 2018). It would be appropriate to discuss the differences here in perspective of that study.

Done.

L278: the two comma's in this sentence should be removed.

Done.

L303-304: I'm missing a clear justification for the assumption that the NO2 lifetime would be 3 hours in the high troposphere. The authors could evaluate this estimate with a model or at least check the literature on what has been done before.

According to literature, the lifetime of NOx in the near field of convection can vary between 3 hours and 2 days [e. g., Pickering et al. (1998), Beirle et al. (2010), Nault et al. (2017)]. The lifetime depends on the height where LNOx is emitted and on how it is transported by convection. Therefore, the lifetime can vary between each particular thunderstorm.

Determining the lifetime of NOx for each studied case would require using a mesoscale model, which is out of the scope of this work. Therefore, we have used different values for the lifetime (see Section 3.3). We have added a statement in the manuscript about the uncertainty of the NOx lifetime and more references showing that its value is still uncertain.

L331: I thought Vtrop,NOx was higher for KNMi than for DLR (because tropospheric slant columns are higher for KNMI)?

Figures 6 to 11 show that there are no significant differences in the tropospheric slant columns of NO2 from TROP-KNMI and TROP-DLR. Differences inVtrop,NOx and Vtropbck are mainly due to differences in VstratNO2. We have added this to the manuscript.

L365: does this mean that non-flashing pixels have probably also been influenced by lightning NOx production? A transport analysis to see where the air in the non0-flashing pixels came from (Barcelona, other regions affected by lightning?) would be useful here.

It does not necessarily mean that non-flashing pixels have been influenced by LNOx. It means that the averaged NOx measured over 3 days with low lightning activity (8, 12 and 13 April 2018) was lower than the averaged background-NOx estimated from the studied cases. We have shown that the background-NOx presents a high variability both during days with low lightning activity (see

Section 2.6) and during the studied cases. Therefore, we concluded in Section 3.2.2 that using fixed values for the background is not adequate in this case-based study.
Please note that we have already included a simple transport analysis by using the winds provided by ERA5 in the upper troposphere to exclude pixels that can have been influenced by LNOx from the estimation of the background-NOx.

L403-404: the conclusion that "differences in LNOx PE using different TROPOMI products are not statistically significant" appears strange to me since the stratospheric corrections are an integral part of the retrieval procedure, and give very different NO2 enhancements.

This statement is based on the T-test performed over the LNOx PE results. As we show in the manuscript, the stratospheric corrections can influence the LNOx estimates. However, according to the performed T-test over all the cases, variability introduced by the stratospheric corrections does not produce statistically significant differences. However, the statistical significance is influenced by the population of the sample. We cannot ensure that the differences would remain statistically insignificant, if the total number of studied cases increased. We have now emphasized this in the manuscript.

References:
Williams, J. E., Boersma, K. F., Le Sager, P., and Verstraeten, W. W.: The high-resolution version of TM5-MP for optimized satellite retrievals: description and validation, Geosci. Model Dev., 10, 721-750, doi:10.5194/gmd-10-721-2017, 2017.

Boersma, K. F., Eskes, H. J., Richter, A., De Smedt, I., Lorente, A., Beirle, S., van Geffen, J. H. G. M., Zara, M., Peters, E., Van Roozendael, M., Wagner, T., Maasakkers, J. D., van der A, R. J., Nightingale, J., De Rudder, A., Irie, H., Pinardi, G., Lambert, J.-C., and Compernolle, S.: Improving algorithms and uncertainty estimates for satellite NO2 retrievals: Results from the Quality Assurance for Essential Climate Variables (QA4ECV) project, Atmos. Meas. Tech., 11, 6651-6678, https://doi.org/10.5194/amt-11-6651-2018, 2018.

Georgoulias, A. K, Boersma, K.F., van Vliet, J., Zhang, X., van der A, R., Zanis, P., and de Laat, J.: Detection of NO2 pollution plumes from individual ships with the TROPOMI/S5P satellite sensor, Environ. Res. Lett., 15 124037, https://doi.org/10.1088/1748-9326/abc445, 2020.

**Reviewer 1**
This paper attempts to estimate the lightning production efficiency of NOx over the Pyrenees and northeastern Spain in Spring 2018 from 3 sources of information: lightning flash counts, TROPOMI satellite above-cloud NO2 column observations, and modelled NOx:NO2 ratios in the upper troposphere. Each of these have considerable uncertainties associated with them. The strength of the work is that the authors address these uncertainties carefully in their approach. Strong about this paper is the use of two different lightning flash networks, two standalone TROPOMI NO2 products, and different approaches to correct for the tropospheric background NO2 not caused by lightning. This is interesting in its own right, as there are considerable uncertainties associated with counting lightning flashes, with the satellite retrieval process, and with our knowledge about tropospheric background NO2. It is also interesting because applying this mini-ensemble provides a robustness check on the quantitative LNOx production, which is at the lower end of previously reported estimates.

The paper has some serious shortcomings which need to be addressed before publication in AMT can be considered in my opinion:

We thank the reviewer for these encouraging comments and for the time spent for the revision.

- A weak point of the paper is the reliance on just one CARIBIC flight (22 June 2005) over the Pyrenees-Ebro area. I understand that there may not have been many flights available, but the representativeness of this alternative method to estimate the NO2 background is debatable. A model (e.g. EMAC) analysis of NOx and NOy in the upper troposphere 22 June 2005 compared to April-May 2018 would be helpful to assess this concern.

  As the reviewer points out, we have used only one CARIBIC flight because there are not many flight available. In particular, commercial flights usually avoid areas with high convection. Thus, we have only found one flight over North of Spain with significant convection (22 June 2005). Despite using only one case, we have obtained a good agreement with previous airborne NO measurements over convective systems without lightning in Europe during the EULINOX campaign [Huntrieser et al. (2002)]. We think this comparison with other measurements in Europe is sufficient to rely on the NO mixing ratio provided by CARIBIC.

- Another major concern is the usage of the concept of the "LNOx air mass factor". In the DOAS-community the AMF is strictly defined as the ratio between the slant and vertical column, and since TROPOMI detects NO2, the use of a lightning NO2 AMF, followed by a model-driven NOx:NO2 correction factor, would be the appropriate way to present this aspect of the approach. It is thus misleading here to use a LNOx AMF since TROPOMI measures NO2, and not NOx. The authors should present the AMF aspect of their approach more clearly, specifically in Figure 5 – the simulations are now input to the center-stage box 'LNOx PE', but in fact the simulations both influence the NO2 AMF (the upper box) and the subsequent NOx-to-NO2 conversion. Also in Eq. (3) the application of an 'AMF_LNOx' is misleading and should be replaced by division by an AMF_LNO2, followed by a correction factor that accounts for the NOx-to-NO2 ratio. To clearly present how the model is required for their ultimate quantification is important given (a) future reproducibility of their results, and (b) the need to prevent leading readers into believing that NOx could somehow be detected from TROPOMI.

  As the reviewer points out, we have used the AMF $NO_x$ to convert the measured SCD $NO_2$ to VCD $NO_x$. However, using an AMF $NO_2$ instead of an AMF $NO_x$ is not appropriate in this case. The convenience of using AMF $NO_x$ instead of AMF $NO_2$ was explained by Beirle et al. (2009):

  "Please note that a two-step conversion (first from NO2
  SCDs into NO2 VCDs using an overall AMF, and then from
  NO2 VCDs into NOx VCDs using a mean NO2/NOx ratio)
  is not appropriate, since both the box-AMFs and the NOx
  partitioning are height dependent, and they do not vary inde-
  pendently because both are particularly influenced by clouds."

  We refer to Beirle et al. (2009) for more details. We have added this explanation to the manuscript for the sake of clarity.

  The use of AMF $NO_x$ instead of AMF $NO_2$ to calculate the VCD $NO_x$ is common in $LNO_x$ PE estimates [e. g.: Bucsela et al. (2013), Pcikering et al. (2016), Bucsela et al. (2019), Allen et al. (2019 & 2021)].

- Figures 6 and 7 actually give little evidence that "areas of high lightning activity coincide with areas with high SCD-NO2". This undermines an important claim of the paper, i.e. that enhanced TROPOMI NO2 can be traced back to previous lightning flashes. The authors should provide more evidence that there is a relationship between flashes and enhanced NO2, e.g. via scatter plots suggesting a spatial correlation for lightning circumstances, and the absence of enhancements on cloudy days without recent lightning activity. The same unfortunately holds for Figures 8 and 9, while the relationship between lightning and enhanced NO2 is more evident of the low-wind day shown in Fig. 10 and 11.

We have calculated the Pearson correlation coefficient (r) between the SCD of $NO_2$ in convective cells with flashes and the total number of flashes reported by ENGLN in each cell averaged over all the studied cases. We have obtained r = 0.18 for TROP-DLR and r = 0.11 for TROP-KNMI. These values indicate a positive correlation between the SCD of $NO_2$ and flashes that is larger for the case of TROP-DLR than for TROP-KNMI. This correlation is larger when we use the tropospheric winds to identify the cells that have been influenced by $LNO_x$. We have copied each flash to the cells that are influenced by the $LNO_x$ produced by the flash with the purpose of calculating the upwind correlation coefficient by taking into account the transport of $LNO_x$. With that we obtain r = 0.20 for TROP-DLR and r = 0.15 for TROP-KNMI. The received larger correlation coefficients indicate that accounting for the transport of $LNO_x$ can improve the estimation of $LNO_x$ PE.

This analysis has been added to the manuscript (Section 3.1).

**Specific comments**

Figure 1 is not particularly helpful. One way to provide more useful context is to overplot the mean NO2 columns on the map. That way the reader can appreciate the difficulty of distinguishing the lightning NO2 signal from the nearby anthropogenic hotspots such as Toulouse, Bordeaux, and Barcelona.

We have removed Fig. 1. Figures 5-10 already show the difficulty of distinguishing the $LNO_2$ signal from the nearby anthropogenic hotspots.

L34: 'estima' --> estimate

Done.

L84: Williams et al. (2017) describes the TM5-MP version which is actually used in the NO2 retrieval. I believe this is a more appropriate reference than the Myriokefalitakis-reference.

Williams, J. E., Boersma, K. F., Le Sager, P., and Verstraeten, W. W.: The high-resolution version of TM5-MP for optimized satellite retrievals: description and validation, Geosci. Model Dev., 10, 721-750, doi:10.5194/gmd-10-721-2017, 2017.

We have added this reference.

L85: where can 'version 2.1_test' be found? Please provide a reference. Are v2.1_test data also available for April and May 2018?

The v2.1 research product is not automatically produced for all the TROPOMI orbits. We produced it on a case-based demand to analyze particular thunderstorms. We have added this clarification to the manuscript.

L108-109: it is unclear here if the authors have imposed temporal coincidence of lightning flashes with the observation of TROPOMI cloud fractions > 0.95 and OCP < threshold. Or has a time window been taken, such as lightning flashes within a few hours of TROPOMI overpass and TROPOMI fulfilling the above cloud criteria? Please clarify.

We have used a 5 hours time window. We have included this clarification in the manuscript.

P6, Figure 3: I think the authors should show here the detection efficiency for April-May 2018 rather than March 2018 – December 2018. After all, the paper is about the lightning NO2 production in April-May 2018.

We have calculated the DE using ISS-LIS lightning data. ISS-LIS is orbiting the Earth in a low Earth orbit. Thus, there are not many coincidences of thunderstorm occurrence and ISS-LIS passage over the particular region. In addition, each point is observed during only 90 seconds during each passage. Therefore, we have included more than one year of data instead of only two months to estimate the DE of ENGLN. In particular, we have found 30 thunderstorms simultaneously detected by ENGLN and ISS-LIS over the area during March 2017 and December 2018. We have added this to the manuscript.

L177: it is unclear how the authors have formulated the SCD here. Which slant column do they mean? The total slant column, which also contains contributions from the stratosphere, or the tropospheric slant column?

Here we mean the SCD $NO_2$ measured by TROPOMI (the total slant column). We have added this to the manuscript.

L180: what is meant with the "absorption of the atmosphere"? I guess this is about the ratio of the slant to vertical (NO2) optical thickness, but it should be clarified.

This is clarified in lines 203-206:
"We use the LNO2 and LNOx vertical profiles from the simulations to calculate the AMFLNOx following Bucsela et al. (2013). We use the TOMRAD forward vector radiative transfer model (Dave, 1965) to calculate the scattering weights for each of the 8 studied cases using the viewing geometry and the cloud properties for each pixel. We obtained AMFLNOx values ranging between 0.28 and 0.71."

We have added a mention of the scattering in these lines.

P9, Figure 4: what was the pressure of the OCP on this day? Please indicate this in the caption. Also indicate the corresponding AMF LNO2 values.

Figure 4 corresponds to a simulation on 13 May 2018. This was the day in May 2018 with the highest $LNO_x$ column density in the simulation. Therefore, we use this case to extract the the mean simulated $LNO_2$ and $LNO_x$ profiles. However, it is important to note that the model is global (2.8 x 2.8 degrees horizontal resolution) and that lightning is parameterized using the updraught as a proxy. Thus, there can be a disagreement between the observed and the simulated total number of flashes in specific thunderstorms.

The total number of lightning flashes detected by EUCLID during this day was not particularly high (165 total flashes). In particular, the total number of flashes 5 hours before TROPOMI overpass was only 29. Therefore, we have not included the analysis of TROPOMI data for 13 May 2018. As a consequence, we cannot provide the value of the OCP nor the AMF for this case. In fact, this would not be useful, as the measurements of TROPOMI on 13 May 2018 are not influenced by fresh $LNO_x$.

P10, Eq. (1): I suggest to include the TROPOMI measurand, i.e. the NO2 column, here. This is now missing from the equation, which may give the impression that TROPOMI somehow provides a tropospheric NOx column, which is not the case.

We have now mentioned that the $V_{tropLNOx}$ is calculated from $TROP-NO_2$ product. The method to calculate it is provided in the following paragraphs.

L274: the authors state that the free tropospheric NO2 "may be overestimated" in TM5-MP, but I see no supporting evidence for this. Do the authors have any reason to suspect this, or could the TM5-MP NO2 background also be underestimated?

We have rephrased. Evaluating the TM5-MP NO2 is out of the scope of this work. We have restricted the discussion to the comparison between the free tropospheric $NO_2$ into the stratosphere provided by STREAM and by TM5-MP models that can influence the estimation of $LNO_x$.

L323 and Tables 1 and 2: the "lower values" of $V_{tropNOx}$ for the DLR vs. KNMI product are unclear to me. Has the $V_{tropbck}$ been subtracted to arrive at $V_{tropNOx}$?

No, the $V_{tropbck}$ is subtracted from the $V_{tropNOx}$ to yield the $V_{tropLNOx}$. The $V_{tropNOx}$ is calculated with eq. (3) using the TROPOMI product and based on the AMF LNOx.

L359: "background NOx …activity" is printed in italics. Not clear why. Same on line 371 and 393-394.

Changed.

P18, Table 3: perhaps useful to also include the overall uncertainty estimate in the table.

Done.

**Reviewer 2**

The authors have analysed several convective systems around the Pyrenees in order to estimate the production of lightning NOx based on TROPOMI NO2 observations.

The Pyrenees are chosen as they are particularly suited for this kind of analysis. While I disagree here (see below), the paper contributes to the scientific question how much NOx is produced by lightning, and it matches the scope of AMT.

The study is generally well written, but there are some inconsistencies and probable bugs in the presented data.

The authors thus have to cautiously check the presented data, correct the existing bugs, and update the LNOx estimates (and, if needed, discussion and conclusion) accordingly.

We thank the reviewer for these encouraging comments and for the time spent for the revision.

In addition, they should consider the additional comments listed below.

**General comments**

1. Study region

As indicated in the title and at several places in the manuscript (e.g. line 70), the Pyrenees are meant to be the focus of this study. In the manuscript, the line of argument is that over the Pyrenees (a) lightning frequency is high, and (b) NOx background is low, and therefore the Pyrenees are an ideal region for this kind of study.

In the presented figures, however, most flashes are observed South or North of the Pyrenees, where also significant boundary layer pollution is present, in particular over large cities like Barcelona, Zaragoza, or Toulouse. This weekens the argumentation considerably, and I am not convinced that the study region is a good choice for studying LNOx from satellite, as the uncertainty of background NOx and the possible uplift of boundary layer pollution severely affect the overall uncertainties.

We agree with the reviewer that there is a significant number of flashes outside the Pyrenees regions. Most of the analyzed thunderstorms extended beyond the Pyrenees, including some areas of the Ebro Valley and Southern France. However, all the studied thunderstorms included a considerable total number of flashes over the Pyrenees. The background-$NO_x$ is a combination of pixels inside and outside the Pyrenees. Including pixels over the Pyrenees contributes to a lower background-$NO_x$ and, as a consequence, to a lower uncertainty in the estimations with respect to studies over polluted areas.

We have clearly shown in the figures and in some points of the manuscript that the studied area is not limited to the Pyrenees. However, we acknowledge that in some lines we suggest that the study is focused on the Pyrenees (as in line 70). This could lead to misunderstandings. We have carefully checked the manuscript and emphasized that adjacent regions are also included in the analysis.

2. Background

As most events are not observed over the Pyrenees, the NOx background might be considerably larger in general. The potential uplift of boundary pollution might bias the observed NO2 over the lightnin pixels.

In addition, also the non-flashing pixels over deep convection might be affected by the advection of LNOx.

As we mentioned above, we agree that the studied cases include a significant number of TROPOMI pixels outside the Pyrenees region. We have now emphasized this in the manuscript.

The inclusion of two different NO2 products is quite illustrative. For instance, the differences in the estimated stratospheric column between the products are far larger than the given uncertainty of the tropospheric background. Please comment on this.

The differences in the estimated stratospheric column between TROP-DLR and TROP-KNMI products and their influence on the LNO$_x$ PE estimates are commented in the manuscript. We have now extended the discussion about these differences.

3. "Lost" NOx

As far as I understood, the study only focusses on cloudy pixels. So in case of LNOx produced within a convective system that disappeared by the time of the TROPOMI overpass, the LNOx would be overlooked in the described procedure, while the respective flashes are included. Consequently, PE is biased low. This seems to actually happen on 29 April.
I propose to skip 29 April, as there are only few TROPOMI observations available, and the extension of the study area to large parts of France is also confusing.

The reviewer is right. As we mention in the manuscript and as other authors indicate in previous studies, the employed method provides an estimation on the fresh-produced LNO$_x$. This is the reason why we only include flashes 5 hours prior to TROPOMI overpass. The averaged age of individual flashes ranges between 0.9 hours for 7 May case and 2.3 hours for 26 May case, while it is 1.9 hours for the 29 April case. We have added this to the mansucript.

We have decided to maintain the 29 April case because the averaged age of individual flashes and the total number of flashes are in the same range as for the other cases. However, we have now commented its particular characteristics in the manuscript.

In addition, this effect has to be included in the discussion and the overall uncertainty estimate.

Effects 2 and 3 go in different directions, are hard to quantify, and will increase the overall PE uncertainty.

We have added to the discussion the possible bias to low LNO$_x$ when using only cloudy pixels. Although indirectly, the effect of the possible overlook of LNO$_x$ is included in the estimation of the uncertainty when using different time windows to count the flashes.

**Inconsistencies and possible bugs**

- Please be consistent with respect to units. Column densities are given in petamolec per cm2 in the text, but in molec per m2 in the figures which makes direct comparisons difficult.

Done.

- Please check tables 1 and 2:
Done, as explained in the following points.
  - Mean Vstrat is quite large (> 8e15 molec/cm2 for DLR), while the maps in Fig. 6ff shows values < 4e15 molec/cm2.

We thank the reviewer for pointing this out. There was an error in the headings of Tables 1 and 2. We do not show the mean VstratNO2 in the tables, but the mean product VstratNO2 x AMFstrat employed in eq. (3) and subtracted to the SCD NO$_2$. We have corrected this error.

  - The 30th percentile of Vtropbck is listed as 9.5e15 molec/cm2 for DLR. This would be a quite considerable tropospheric pollution. Please check.

There is a typo. The Vtropbck is 9.5e14 molec/cm2 instead of 9.5e15 molec/m2.

- Please check Figs. 10 and 11. The number of flashes displayed in the figures is considerably different, while the numbers listed in Tables 1 and 2 are almost the same for 28 May. Please also check for the other days.

We have checked that the numbers are correct. As the maps show small points, the visualization of the total number of flashes is not evident.

**Minor comments**

Line 8: Over the Pyrenees, the NO2 background is low indeed. But for the investigated events over the Ebro valley, this is not the case, see e.g. http://www.tropomi.eu/data-products/nitrogen-dioxide

We have now commented in the manuscript that pixels over polluted areas can influence the background-$NO_x$ in our study.

Lines 31-34: "Nadir-viewing satellite instruments ... estimate the column densities of NO2 over thunderstorms" - I think this is too fuzzy. The satellite instruments measure spectra. This allows quite accurate quantification of NO2 slant column densities. The conversion into vertical column densities is the main challenge, involving further input data (like cloud fraction and surface albedo, so this might be denoted as an "estimate". But usually, the retrieval focusses on cloud free conditions. Focussing on thunderstorm clouds instead is a quite different and challenging setup which should be pointed out here.

We have rephrased.

Lines 45-46: I propose to flip the order: "The horizontal resolution at nadir is 3.6 km × 7.2 km before 6 August 2019, while it is 3.6 km × 5.6 km thereafter."

Done.

Figure 1: I propose to show a map of tropospheric NO2 instead in order to assess the "clean background" issue.

We have removed this figure. The rest of the figures already show the map of tropospheric $NO_2$.

Line 70: "8 deep convective systems in the Pyrenees": The number of actual flashes over the Pyrenees is quite low. The presented events reveal highest lightning activity South and North of the Pyrenees, with several anthropogenic sources present (e.g. Barcelona, Toulouse).

We have now commented that other regions are also included.

Section 2.3: I think it is quite bold to estimate "the" tropospheric background NOx column (!) from one day of aircraft measurements (in-situ measurements at 12 km!). The authors should discuss the limitations and uncertainties of this approach in more detail and compare with model results.

We have extensively commented on this issue in the answer to the reviewer 1 and introduced some changes in the manuscript. As the discussion is public, we refer to the answers to reviewer 1.

Eq. 1: Please specify which quantities are dependent on space (TROPOMI pixel) and which quantities are just scalar (means, percentiles). For this you might use Copernicus style where vectors and matrices are shown in bold type. Please also explain how the total PE is calculated (spatial mean or summation over which area/pixels?).

We have now used the Copernicus style for the vectors and matrices. The reader can now follow the reasoning and the following equations to better understand how the PE is calculated. The PE is influenced by scalars, spatial mean and spatial medians, as can be seen in eq. (1-3).

Eq. 3: Stratospheric VCD and AMF are provided for each TROPOMI pixel. So the stratospheric correction can be performed for each pixel as well. I see no reason to calculate the average here.

We are following the same approach as Allen et al. (2021), who used the average instead of individual pixels. The reason is explained by Allen et al. (2021) when referring to their Eq. (5) including the average:

"...TROPOMI $NO_2$ product were often missing values for VtropNO2 and sometimes missing values for VstratNO2 and AMFstrat in regions affected by deep convection and lightning. The use of Equation 5 increased the number of pixels available to estimate VtropLNOx and allowed for more robust estimates of PE".

As one of the goals of the present study was to establish a comparison with previous estimates of LNOx over the US by Allen et al. (2021), we have applied the same method.

We have now included this explanation in the manuscript.

Lines 227-231: Several events are outside the Pyrenees, with considerably higher background NOx. In addition, in case of deep convection, tropospheric pollution might be uplifted and transported over considerable distances. So the local tropospheric background estimate over the clean Pyrenees can at best be considered as a lower limit.

We have added parts of this comment to the manuscript. The background estimate over the clean Pyrenees is calculated for days with convection.

Line 251: Please explain why the DLR product is missing.

Files in the database are missing, but the reason is not known.

Line 509: Please list the existing and the new methods here.

We have rephrased.

- Figs. 6ff: Maps of stratospheric AMF and VCD are not that informative. Listing these values in tables 1&2 would be sufficient, but the numbers listed there need to be checked.
Instead, I would like to see maps of AMF_LNOx here.

We have now removed the map of stratospheric AMF and introduced the map for AMF_LNOx. We have maintained the map of stratospheric VCD because the size of the figure does not change when removing only one panel.

- Fig. 12 needs some further explanations:

  - how was the median calculated?

The average values shown for each point are calculated over the values provided by ERA5 on the pressure levels 200 hPa, 250 hPa, 300 hPa, 350 hPa, 400 hPa, 450 hPa and 500 hPa. The median values shown in the title of the figure are the spatial median over the average in each pixel. We have introduced this in the manuscript.

- what is the meaning of color?

The color is related with the wind velocity. We have added a colorbar and changed the color to make it more visual.

---

## Author Response (AR2)

We thank the reviewer for the time spent in the revision of this manuscript.

The authors have adressed most comments raised by the reviewers.
There are some remaining issues that should be fixed/clarified before publication:

- Background Caribic:
In their reply to reviewer 1, the authors state that "We think this comparison
with other measurements in Europe is sufficient to rely on the NO mixing ratio provided by
CARIBIC."
I don't see this statement supported by the data presented. It is still a single mixing ratio measured
on one day, used to construct a full tropospheric column. Thus it has a quite large uncertainty. The
authors should state that clearly in the revised manuscript.

In addition, in reply to reviewer 1 the authors state that the background column derived from
CARIBIC differs for KNMI and DLR due to different assumptions made. This important
information is missing in the manuscript.

Measurements in convective systems are usually done during flight campaigns. In this case, there
are no data for chemical measurements in the studied area apart from the measurements provided by
CARIBIC. In addition, the measured mixing ratio of NO was in agreement with previous airborne
NO measurements over convective systems without lightning in Europe during the EULINOX
campaign. We have added more details in the manuscript.

In replies to reviewer 1, we have stated that the $LNO_x$ estimated from the CARIBIC-bakcground is
different from that calculated from TROP-KNMI and TROP-DLR bakcground. The $LNO_x$ from
TROP-KNMI and TROP-DLR is calculated as the average within all the cases showed in Tables 1
and 2. However, the background- $NO_x$ calculated from CARIBIC ($0.75 \times 10^{19}$ molec m$^{-2}$) is clearly
within the range of the background-$NO_x$ showed in Tables 1 and 2 (between negative values and 3.9
$\times 10^{19}$ molec m$^{-2}$).

- Lifetime:
In some papers, the lifetime of NOx in the upper troposphere is stated to be of the order of even
several days (up to >10), e.g. Penner et al., JGR, 1998.
The authors choose a far shorter value of few hours. This is not comprehensible just from the range
covered by literature values.
It only becomes comprehensible after reading Nault et al. who provide an updated estimate and
state that previous values are biased high because they do not consider reactions that are significant
for lightning NOx.
As this is important for the current study, the authors should expand the discussion of lifetimes and
should shortly summarize the results from Nault et al. and explain, why the older (longer) lifetimes
are probably biased high.

Done. We have added that the lifetime can vary between 2 h and several days according to the
literature. We have explained the essential features of the new estimations reported by Nault et al.
(2017).

- Missing orbit:
The reply that "Files in the database are missing, but the reason is not known." is quite
unsatisfactory.
Since several Co-authors of this paper are involved in the algorithm development and data

processing for TROPOMI, I would like to see a more concrete reason what exactly is missing that causes a gap in the DLR product, while the KNMI product is available.

At the time that the specific TROPOMI NO2 and cloud data was provided by DLR and KNMI for the $LNO_x$ study, the TROPOMI cloud data from DLR was not available for one orbit. This was due to a technical processing issue, which could unfortunately not be solved in time for the $LNO_x$ study.

---

## Author Response (AR3)

We thank the editor for the time spent in the revision of this manuscript.

On line 186, you give one number for the background column estimated from CARIBIC. In the replies to the reviewers, however, you provide too different numbers (KNMI vs. DLR based). These two different numbers should also be listed in the manuscript around line 186, as asked by the latest reviewer comments.

We have added the values in line 186.

Remarks from the preceding review file validation:

For the next revision, I kindly ask you: 1. delete the section "Copyright statement" from the *.pdf supplement file. 2. Please keep colour blindness in mind and avoid the parallel usage of green and red. For a list of colour scales that are illegible to a significant number of readers, please visit ColorBrewer 2.0 (http://www.colorbrewer2.org/).

We have deleted the section "Copyright statement" from the *.pdf supplement file. We have checked that the figures are as colour blindness as possible. We have used symbols (dots, dashed lines, etc) in figure containing green and red.